# UniAdapter: Unified Parameter-Efficient Transfer Learning for Cross-modal Modeling

**Haoyu Lu**[1]  **Yuqi Huo**[3]  **Guoxing Yang**[1]  **Zhiwu Lu**[1,*]**Wei Zhan**[2]  **Masayoshi Tomizuka**[2]  **Mingyu Ding**[2,*]

[1]Gaoling School of Artificial Intelligence, Renmin University of China, Beijing, China
[2]University of California, Berkeley, United States
[3]Baichuan Inc.

```
{lhy1998, luzhiwu}@ruc.edu.cn     myding@berkeley.edu
```

## Abstract

Large-scale vision-language pre-trained models have shown promising transferability to various downstream tasks. As the size of these foundation models and the number of downstream tasks grow, the conventional full fine-tuning paradigm becomes impractical due to heavy computational and storage costs. This paper proposes UniAdapter, which unifies unimodal and multimodal adapters for parameter-efficient cross-modal adaptation on pre-trained vision-language models. Specifically, adapters are distributed to different modalities and their interactions, with the total number of tunable parameters reduced by partial weight sharing. The unified and knowledge-sharing design enables efficient adaptation to various downstream tasks with powerful cross-modal representations, requiring only 1.0%–2.0% tunable parameters of the pre-trained model. Extensive experiments on 7 cross-modal downstream benchmarks (including video-text retrieval, image-text retrieval, VideoQA, VQA and caption) show that in most cases, UniAdapter not only outperforms the state-of-the-arts, but even surpasses the full fine-tuning strategy. Notably, on the MSRVTT retrieval task, UniAdapter achieves 49.7% recall@1 with only 2.2% tunable model parameters, outperforming the latest competitors by 2.0%. The code and models are available at https://github.com/RERV/UniAdapter.

## 1 Introduction

The pretrain-finetune paradigm has achieved great success in natural language processing (NLP) (Devlin et al., 2019; Ding et al., 2023), computer vision (CV) (Wang et al., 2022c), and multimodal modeling (Radford et al., 2021; Jia et al., 2021), where models are first pre-trained with large-scale data, and then fully fine-tuned for each downstream task. Recent research further finds that fine-tuning/adapting a foundation model to a new modality by introducing additional trainable modules significantly outperforms previous works, such as temporal modeling modules (Gao et al., 2021; Ju et al., 2022; Lu et al., 2022) for image-to-video transferring (see Figure 1 (a)).

However, as foundation models become increasingly large (Alayrac et al., 2022; Touvron et al., 2023; OpenAI, 2023; Anil et al., 2023; Chen et al., 2023) and the number of downstream tasks increases, particularly in multimodal scenarios, the traditional method of full fine-tuning becomes impractical due to the significant computational and storage requirements that it entails. Finding new ways to efficiently transfer foundation models to downstream tasks without incurring excessive costs, becomes an important challenge in the field.

Alternative approaches have been explored to address the above challenge. A straightforward approach is the use of Linear Probe, which freezes almost the entire model and only tunes a lightweight head for each task. It is only sub-optimal since the representation and the feature space are fixed. Another line of research alleviates the problem by few-shot learning with very large extra modules added to the foundation models (e.g., Flamingo (Alayrac et al., 2022)), which is still far from the full fine-tuning strategy. Recently, parameter-efficient adapters show remarkable results to generalize foundation models in many research fields. In NLP and CV, tunable efficient adapters (Houlsby et al., 2019; Hu et al., 2022b) and tunable prompt vectors (Li & Liang, 2021) are applied with a frozen backbone during transfer learning (Houlsby et al., 2019; Hu et al., 2022b). They also show great potential for cross-modal modeling (Ju et al., 2022; Sung et al., 2022; Pan et al., 2022; Gao et al., 2023), as

---

*Corresponding authors.

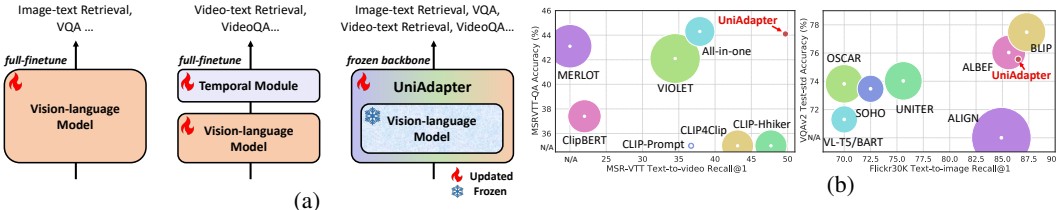

Figure 1: (a) Comparison between standard vision-language full fine-tuning paradigm (for image-text and video-text downstream tasks) and our parameter-efficient approach for various downstream tasks. (b) Performance comparison on cross-modal retrieval (horizontal axis) and visual question answering (VQA, vertical axis) tasks. The bubble size denotes the total tunable parameters. On either dataset, our UniAdapter achieves competitive performance on both two tasks while enjoying significantly fewer tunable parameters. N/A: several works focus on either VQA or cross-modal retrieval, and we set N/A for the non-report results.

they enable the transfer of pre-trained foundation models from cross-modal to single-modal (*e.g.*, video classification (Pan et al., 2022; Gao et al., 2023)) or other downstream tasks (*e.g.*, image-text reasoning (Sung et al., 2022)). However, the above works typically consider either a single modality or a single downstream task, without the support of single-/cross-modal and different downstream tasks. Considering various downstream tasks in multimodal modeling (*e.g.*, video-text retrieval, image-text retrieval, video and visual question answering), a unified representation of adapters applicable to different multimodal downstream tasks is crucial. Meanwhile, previous approaches typically apply adapters without considering cross-modal interaction and knowledge sharing between them, which is the key to cross-modal modeling.

Motivated by the above observations, in this work, we investigate a critical problem of ***efficiently transferring a vision-language model to unified cross-modal modeling***, which aims to enable a vision-language pre-training model to adapt to unified modalities (*e.g.*, image and video) as well as unified cross-modal downstream tasks (*e.g.*, retrieval and reasoning) in a parameter-efficient principle. We propose UniAdapter, which unifies adapters for multimodal modeling and distributes them to each modality and cross-modal interaction. UniAdapter has several appealing benefits that previous works do not have: **1)** To model the cross-modal interactions, we introduce a knowledge-sharing scheme, where the down-projection layer in all adapters is shared while the up-projection can learn modality-specific knowledge. **2)** To preserve the integrity of language queries during the cross-attention process in multimodal models, we incorporate residual learning for language queries. **3)** We propose parameter-free frame-aware attention to unify the video and image modalities with no cost, not only making our approach applicable to more downstream tasks, but also alleviating the noise issue in videos. With these design considerations, our UniAdapter capitalizes a pre-trained vision-language model for unified cross-modal downstream tasks by introducing a few tunable parameters.

Our contribution is threefold: **1)** We investigate the problem of unified parameter-efficient cross-modal transfer learning, which allows for the efficient utilization of a pre-trained vision-language model for a range of cross-modal downstream tasks. **2)** We propose UniAdapter, a simple, efficient, yet effective framework with carefully considered designs, such as knowledge sharing and query residuals. To our best knowledge, we are the first adapter-based work that is applicable to various downstream tasks (including retrieval and reasoning) from both image-language and video-language domains. **3)** Extensive evaluations on six cross-modal downstream benchmarks show that our UniAdapter generally outperforms previous arts with fewer parameters, especially in the video domain.

## 2 RELATED WORK

**Parameter-efficient Transfer Learning.** Parameter-efficient Transfer Learning technologies (Houlsby et al., 2019; Hu et al., 2022b; Ding et al., 2023) are first proposed in the NLP domain to alleviate the heavy training and storage cost in the full fine-tuning process facing the increasing foundation model size. These approaches aim to adapt a frozen large-scale model to downstream tasks by introducing small updating parameters. Recent works (Pan et al., 2022; Chen et al., 2022) also validate its effectiveness in the CV domain. Nevertheless, cross-modal parameter-efficient transfer learning is still not well explored. Although several pioneer works (Pan et al., 2022; Sung et al., 2022; Gao et al., 2023) are proposed for efficient cross-modal modeling, these works

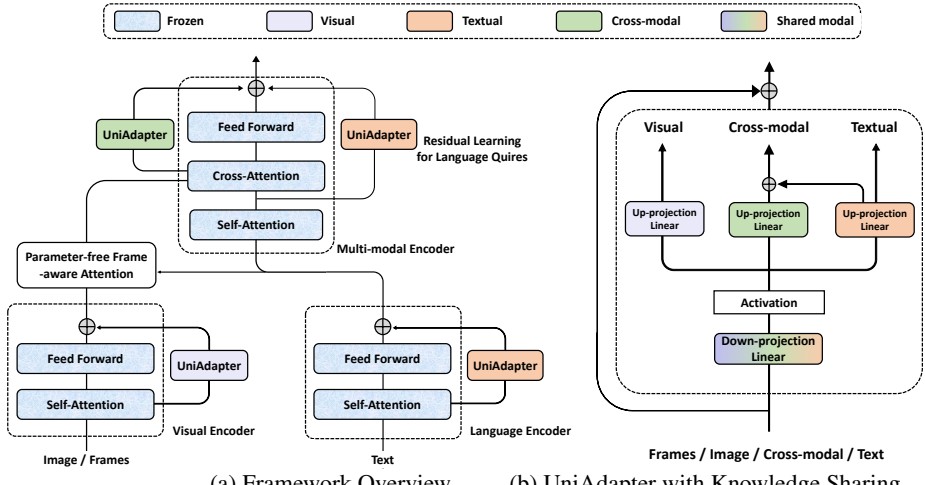

(a) Framework Overview     (b) UniAdapter with Knowledge Sharing

Figure 2: (a) Semantic illustration of our overall parameter-efficient transfer learning framework. UniAdapters with the same color share the same weight. (b) Detailed design of our UniAdapter. Each modality shares a unified down-projection layer. The cross-modal up-projection branch considers utilizing knowledge from the textual up-projection layer to better learn the fusion information.

typically directly apply standard adapter or prompt approaches and focus on either single-modality tasks (*e.g.*, visual classification (Gao et al., 2023)) or single-type downstream tasks (*e.g.*, image-text reasoning (Sung et al., 2022)). In this work, our proposed UniAdapter is different and more general, which unifies unimodal and multimodal adapters for parameter-efficient cross-modal modeling, and can cope with unified modalities and various downstream tasks.

**Vision-language Modeling.** Video-language models can be roughly divided into two-stream models (*e.g.*, CLIP (Radford et al., 2021), ALIGN (Jia et al., 2021)) and single-stream models (*e.g.*, SimVLM (Wang et al., 2022d), OSCAR (Li et al., 2020)). Recently methods (*e.g.*, ALBEF (Li et al., 2021), BLIP (Li et al., 2022), BLIP2 (Li et al., 2023a), BEIT 3 (Wang et al., 2023)) combine the advantages of both encoder-based and decoder-based methods, thus can support both cross-modal alignment tasks and multimodal generation tasks in one foundation model. While these image-text foundation models are extendable to various downstream tasks, the growing size of backbones (such as Flamingo (Alayrac et al., 2022), Qwen-VL (Bai et al., 2023), PaLI-X (Chen et al., 2023)) increases the burden of training and storage requirements.

## 3 METHODOLOGY

In this section, we first briefly describe the vision-language framework and the standard adapter. We then introduce our UniAdapter with query-based residual learning and frame-aware attention, to show how we capitalize a large-scale vision-language model for a wide range of downstream tasks from both image-language and video-language domains. The overall architecture is illustrated in Figure 2.

### 3.1 PRELIMINARY

**Vision-language Framework.** We utilize a hybrid-stream architecture as our frozen backbone for it combines the advantages of both two-stream and single-stream methods with superior performance and relatively high inference speed, which consists of a visual encoder (ViT (Dosovitskiy et al., 2021)), a language encoder (BERT (Devlin et al., 2019), and a multimodal encoder as shown in Figure 2.

Given an image/video-text pair, our model first utilizes the unimodal encoders to extract the visual features $\mathbf{f}^v = \{f^v_{\mathrm{CLS}}, f^v_0, f^v_1, ...\}$ and the textual features $\mathbf{f}^t = \{f^t_{\mathrm{CLS}}, f^t_0, f^t_1, ...\}$, where $f^v_{\mathrm{CLS}}$ and $f^t_{\mathrm{CLS}}$ are [CLS] tokens. The cross-modal contrastive objectives are then applied for instance-level alignment on [CLS] tokens. The extracted visual features $\mathbf{f}^v$ and textual features $\mathbf{f}^t$ are then fed into the multimodal encoder for cross-modal token-level modeling. Specifically, the multimodal encoder takes text features as input and the visual features are inserted into each cross-attention layer for injecting the visual features. For video-language domain tasks, we first utilize the visual encoder to

extract each frame feature $\mathbf{f}^e = \{f^e_{\text{CLS}}, f^e_0, f^e_1, ...\}$. Then we concatenate the frame features as the visual input for each cross-attention layer.

**Adapter.** Adapter (Houlsby et al., 2019) is proposed for parameter-efficient transfer learning in the NLP domain, which freezes the pre-trained parameters and inserts small tunable modules between each layer. Each adapter consists of a down-projection layer $W_{down} \in \mathcal{R}^{(d \times r)}$, a nonlinear activation function $\sigma$ and an up-projection layer $W_{up} \in \mathcal{R}^{(r \times d)}$, where $d$ (or $r$) is the input (or bottleneck) dimension. Given an input feature $x \in \mathcal{R}^d$, the computation process can be given in a residual from:

$$Adapter(x) = x + s \cdot \sigma(xW_{down})W_{up}, \tag{1}$$

where $s$ is the scaling factor.

### 3.2 OVERALL ARCHITECTURE

UniAdapter aims to enable a pre-trained vision-language model for unified cross-modal downstream tasks in a parameter-efficient principle. Apart from that we evenly insert uniadapters into each transformer layer of textual, visual, and multimodal encoders as shown in Figure 2(a), our framework has three unique designs for cross-modal transfer learning: (1) To preserve the integrity of language queries during the cross-attention process in multimodal encoders, we incorporate residual learning for language queries. (2) We introduce unified and cross-modal knowledge-sharing designs, where the down-projection layer in all adapters is shared while the up-projection can learn modality-specific knowledge, as shown in Figure 2(b). (3) Considering the noisy issue in video frames, we propose parameter-free frame-aware attention to unify the video and image modalities with no cost and alleviate the noisy problem that exists in video-language domains. We discuss each part below.

### 3.3 RESIDUAL LEARNING FOR LANGUAGE QUERIES

The multimodal encoder is adopted for cross-modal token-level modeling, which takes text features as query input and the visual features are inserted into each cross-attention layer for injecting the visual features. Standard approaches insert adapters behind the multi-head attention in the transformer encoder architecture. Nevertheless, directly following this approach (inserting adapters behind the cross-attention layer) for the multimodal encoder is hard to deal with hybrid information, and may break the integrity of language queries during the cross-attention process in the multimodal encoder. Therefore, we introduce Residual Learning for Language Queries to address this issue.

Specifically, each multimodal encoder block consists of a multi-head self-attention (MSA), a multi-head cross-attention (MCA), and a fully connected feed-forward network (FFN). The multimodal encoder takes text features $\mathbf{f}^t$ as input and the visual features are inserted into each cross-attention layer for injecting the visual features as shown in Figure 2. Each cross-attention layer takes the self-attention output features $\boldsymbol{q}$ as query $Q$ and the visual features $\mathbf{f}^v$ as key $K$ and value $V$. The computation process of each block can be formulated as:

$$
\begin{aligned}
\boldsymbol{q} &= \boldsymbol{l_{l-1}} + \text{MSA}(\boldsymbol{l_{l-1}}), \\
\boldsymbol{h} &= \boldsymbol{q} + \text{MCA}(Q = \boldsymbol{q}, K = \mathbf{f}^v, V = \mathbf{f}^v), \\
\boldsymbol{l_l} &= \boldsymbol{h} + \text{FFN}(\boldsymbol{h}),
\end{aligned}
\tag{2}
$$

where $\boldsymbol{l_0} = \mathbf{f}^t$, and $\boldsymbol{l_l}$ denotes the output features of the $l$-th layer. When the standard Adapter is inserted behind the cross-attention layer, Equation 2 can be rewritten as:

$$\boldsymbol{l_l} = Adapter(\boldsymbol{h}) + \text{FFN}(\text{LN}(\boldsymbol{h})), \tag{3}$$

where LN denotes the layer norm. It can be observed from Equation 2 that, the hidden state $\boldsymbol{h}$ contains both Query features as well as cross-modal fusion features. Learning such hybrid information with a single modality adapter is very hard. Moreover, the textual query information may be lost during transmission in each cross-encoder block. Therefore, we propose to capture/maintain the Query information by introducing an additional adapter in a residual form, termed Query-Residual adapter. Specifically, we insert it behind the self-attention layer and directly add the output to the feed-forward layer in a residual form, as shown in Figure 2. Equation 2 can now be rewritten as:

$$\boldsymbol{l_l} = Adapter(\boldsymbol{q}) + Adapter(\boldsymbol{h}) + \text{FFN}(\text{LN}(\boldsymbol{h})). \tag{4}$$

Simply introducing the Query-residual adapter may bring extra updating parameters, which is not expected in our lightweight principle. We observe that the textual encoder also takes the textual

features as input and adds the output to the feed-forward layer. Therefore, the textual adapter knowledge can be shared with Query-Residual adapter by fully weight-sharing between the two adapters to avoid extra parameter costs. We find that this weight-sharing mechanism even brings better performance, as shown in Appendix Table 1.

### 3.4 UNIADAPTER

To transfer a vision-language model to downstream tasks, a straightforward way is to inject adapters into each modality module (visual, textual, and multimodal fusion). However, utilizing separate adapters for each modality brings relatively high parameters. Meanwhile, there are no cross-modal interactions among these adapters, leading to suboptimal performance. We propose UniAdapter to address the above issues, which unifies unimodal and multimodal adapters into one framework by partial weight sharing.

The core idea of UniAdapter is to share the knowledge from multiple modalities to enhance cross-modal interaction meanwhile reducing extra tunable parameters. As illustrated in Figure 2(b), UniAdapter consists of a unified down-projection layer $W_{down} \in \mathcal{R}^{(d \times r)}$, a nonlinear activation function $\sigma$, and a modality-specific up-projection layer $W_{up}^{\mathcal{M}} \in \mathcal{R}^{(r \times d)}, \mathcal{M} \in \{\mathcal{V}, \mathcal{T}, \mathcal{C}\}$, where $d$ and $r$ are the input and bottleneck dimensions, and $\mathcal{V}, \mathcal{T}, \mathcal{C}$ denote the visual, textual and cross-modal modality, respectively. The down-projection layer in all UniAdapters is shared while the up-projection can learn modality-specific knowledge. Below we introduce UniAdapter for each modality.

**Unimodal Case.** Although we apply a unified down-projection for cross-modal knowledge-sharing, learning modality-specific representation is also important for the unimodal encoders. Therefore, we apply two modality-specific up-projection layers $(W_{up}^{\mathcal{V}}, W_{up}^{\mathcal{T}})$ respectively for the visual encoder and textual encoder:

$$UniAdapter(x^{\mathcal{V}}) = x^{\mathcal{V}} + s \cdot \sigma(x^{\mathcal{V}} W_{down}) W_{up}^{\mathcal{V}},$$
$$UniAdapter(x^{\mathcal{T}}) = x^{\mathcal{T}} + s \cdot \sigma(x^{\mathcal{T}} W_{down}) W_{up}^{\mathcal{T}}, \tag{5}$$

where $s$ denotes the scaling factor, and $x^{\mathcal{V}}$ and $x^{\mathcal{T}}$ denote the visual and textual features, respectively.

The visual encoder and textual encoder take the same transformer encoder architecture and we follow MAM (He et al., 2022) to inject UniAdapter between the self-attention layer and the feed-forward layer (see Figure 2).

**Cross-modal Case.** We also utilize a specific up-projection layer for multimodal encoder transfer learning. However, as we mentioned in Sec. 3.3, the input features consist of the Query features as well as cross-modal fusion features. Learning such hybrid information with a single adapter is very hard. Following the design of Sec. 3.3, we consider reusing the textual up-projection layer $W_{up}^{\mathcal{T}}$ into UniAdapter for capturing the textual information. In this way, the cross-modal up-projection layer $W_{up}^{\mathcal{C}}$ can cope with the cross-modal information more easily. The UniAdapter on the cross-modal modality can be expressed as:

$$UniAdapter(x^{\mathcal{C}}) = x^{\mathcal{C}} + s \cdot \left[ \sigma(x^{\mathcal{C}} W_{down}) W_{up}^{\mathcal{T}} + \sigma(x^{\mathcal{C}} W_{down}) W_{up}^{\mathcal{C}} \right],$$

where $s$ denotes the scaling factor (as in the unimodal case), and $x^{\mathcal{C}}$ denotes the cross-modal features.

For multimodal encoder, we insert UniAdapter between cross-attention and feed-forward layer. We additionally consider Query-residual Adaption introduced in Sec. 3.3. Equation 2 can be rewritten as:

$$\boldsymbol{l_l} = UniAdapter(\boldsymbol{q}) + UniAdapter(\boldsymbol{h}) + \text{FFN}(\text{LN}(\boldsymbol{h})). \tag{6}$$

### 3.5 PARAMETER-FREE FRAME-AWARE ATTENTION

For video downstream tasks, we concatenate the frame features extracted from the visual encoder as the visual input for video-level cross-modal alignment. However, this approach considers all frames with equal weight and ignores the noise and misalignment problem in videos. Inspired by LGDN (Lu et al., 2022), we propose Parameter-free Frame-aware Attention (PFA) for video-text retrieval, which highlights the tokens in salient frames and while suppressing tokens in noisy or irrelevant frames during the cross-attention process, without introducing extra parameters.

Formally, given a video-text pair with the extracted frame features $\{f_{\text{CLS},i}^e, f_{i,j}^e | i = 1, ..., n, j = 1, ..., m\}$, where $n$ is the length of the video and $m$ is the length of the token sequence, we first

identify the attention weight $A_i$ of the $i$-th frame by computing the dot-product of the frame features and the paired text [CLS] token features $f_{\text{CLS}}^t$:

$$A_i = \frac{exp(f_{\text{CLS},i}^e \cdot f_{\text{CLS}}^t)}{\sum_i exp(f_{\text{CLS},i}^e \cdot f_{\text{CLS}}^t)}, \quad (7)$$

Then we apply the PFA attention weight for each frame feature $\mathbf{f}^e$ to formulate the final input visual features, which can be formulated as:

$$PFA(\mathbf{f}_i^e) = \{f_{\text{CLS},i}^e, A_i * f_{i,j}^e | 1 \leq i \leq n, 1 \leq j \leq m\}. \quad (8)$$

## 4 EXPERIMENT

### 4.1 DATASETS AND SETTINGS

**Downstream Datasets.** We evaluate our proposed UniAdapter on 7 downstream datasets, including video-text retrieval datasets: MSR-VTT (Xu et al., 2016) and DiDeMo (Hendricks et al., 2017); image-text retrieval datasets: MSCOCO (Lin et al., 2014) and Flickr30K (Plummer et al., 2015); video question answering dataset: MSRVTT-QA (Xu et al., 2017); visual question answering dataset: VQAv2 (Goyal et al., 2017); and Caption dataset: MSCOCO (Lin et al., 2014). We present the details of the downstream datasets as well as the evaluation metrics in Appendix Section 4 & 5.

**Implementation Details.** We apply BLIP-base (Li et al., 2022) as our vision-language backbone for both image-text and video-text downstream tasks (also explore larger and other backbones in Appendix A). The parameters of the BLIP model are kept frozen during the fine-tuning process. We set the UniAdapter hyper-parameters uniformly for all modalities as: input/output dimension $d =$

Table 1: Inserting Adapter (r=512) into different modalities (Visual $\mathcal{V}$, Textual $\mathcal{T}$ and Cross-modal $\mathcal{C}$) on Didemo. # Tunable: the number of tunable parameters.

| $\mathcal{V}$ | $\mathcal{T}$ | $\mathcal{C}$ | # Tunable | R@1 | R@5 | R@10 | R@Mean | MedR |
|---|---|---|---|---|---|---|---|---|
| ✓ | | | 9.5M | 42.6 | 70.9 | 79.4 | 64.3 | 2.0 |
| | ✓ | | 9.5M | 40.0 | 64.6 | 74.7 | 59.8 | 2.0 |
| ✓ | ✓ | | 19.0M | 44.5 | 73.1 | 80.9 | 66.2 | 2.0 |
| | | ✓ | 9.5M | 47.7 | 73.4 | 82.8 | 68.0 | 2.0 |
| ✓ | ✓ | ✓ | 28.4M | **49.9** | **76.2** | **83.0** | **69.7** | 2.0 |

768, bottleneck dimension $r = 128$ (4.8M) or $r = 512$ (19.0M), and scaling factor $s = 0.1$. Following previous works, we initialize the weights of down-projection layers for UniAdapter with Kaiming Normal (He et al., 2015) and configure the weights of the up-projection layers with zero initialization.

For video-text downstream tasks, we uniformly sample $N = 8$ frames per video during training, $N = 16$ frames per video during inference (but $N = 8$ for ablation study). All experiments are conducted on 8x NVIDIA 3090Ti (24G) GPUs. More details are given in Appendix.

### 4.2 ABLATION STUDY AND ANALYSIS

In this subsection, we conduct comprehensive ablation studies to reveal how to successfully build a parameter-efficient transfer learning framework for cross-modal modeling and investigate the contributions of different components of our UniAdapter. If not specifically indicated, we set bottleneck dimension $r = 512$ for Adapter/UniAdapter, inference frames $N = 8$ as the default setting in the ablation study.

**Where to Insert Adapter.** Prior approaches (Houlsby et al., 2019; Chen et al., 2022) have successfully applied adapters in single-modality domains. To replicate the success, we choose to first identify which modality module is more crucial for cross-modal transfer learning. Specifically, we first apply adapters for different modality encoders as shown in Table 1. It can be observed that: (1) Inserting adapter into multimodal encoder significantly outperforms visual or textual modality (and even both visual and textual modality), suggesting that multimodal adaption should be paid more attention to. (2) Inserting adapters for all modality encoders achieves the best performance, and hence we adopt adapters for all modality modules as our default setting.

**Effectiveness of Each Component.** We compare our UniAdapter with three baselines (Linear Probe, Full fine-tuning, and Adapter) and demonstrate the contribution of each component of our UniAdapter in Table 2. Note that we start with the standard Adapter, which evenly inserts adapters into all

Table 2: Ablation study results for the proposed components on the Didemo test set and MSRVTT-QA valid set. # Tunable: the number of tunable parameters. PFA: Parameter-free Frame-aware Attention. The second best result is marked by underline.

| Method | # Tunable | Didemo Text-to-Video Retrieval | | | | | MSRVTT-QA |
| | | R@1 | R@5 | R@10 | R@Mean | MdR | Val Acc |
|---|---|---|---|---|---|---|---|
| Linear Probe | 0.4M | 39.7 | 64.6 | 74.9 | 59.7 | 2.0 | - |
| Full fine-tuning | 223 / 337M | 51.3 | **79.1** | **85.7** | **72.0** | **1.0** | 43.0 |
| Adapter (r=128) | 7.1 / 4.8M | 47.4 | 73.5 | 81.4 | 67.4 | 2.0 | 41.5 |
| Adapter (r=512) | 28.4 / 19.0M | 49.6 | 75.9 | 82.8 | 69.4 | 2.0 | 42.8 |
| UniAdapter (r=128) | 4.8M | 49.0 | 75.5 | 83.3 | 69.3 | 2.0 | 43.6 |
| -Weight-sharing | 7.1M | 49.7 | 75.5 | 83.4 | 69.5 | 2.0 | 42.1 |
| -Query-residual Adaption | 7.1 / 4.8M | 48.1 | 74.2 | 82.4 | 68.2 | 2.0 | 41.5 |
| -PFA | 7.1 / 4.8M | 47.4 | 73.5 | 81.4 | 67.4 | 2.0 | - |
| UniAdapter (r=256) | 19.0M | **52.1** | 77.3 | 85.2 | 71.5 | **1.0** | **44.5** |
| -Weight-sharing | 28.4M | 51.3 | 76.5 | 84.2 | 70.7 | **1.0** | 43.7 |
| -Query-residual Adaption | 28.4 / 19.0M | 50.1 | 76.1 | 83.5 | 69.9 | **1.0** | 42.8 |
| -PFA | 28.4 / 19.0M | 49.6 | 75.9 | 82.8 | 69.7 | 2.0 | - |

Table 3: Comparative results obtained by different weight-sharing strategies used for UniAdapter on the Didemo test set and MSRVTT-QA valid set. # Tunable: the number of tunable parameters. The second best result is marked by underline.

| Method | # Tunable | Didemo Text-to-Video Retrieval | | | | | MSRVTT-QA |
| | | R@1 | R@5 | R@10 | R@Mean | MdR | Val Acc |
|---|---|---|---|---|---|---|---|
| w/o Share (r=512) | 28.4M | **52.4** | **77.6** | 84.2 | 71.4 | 1.0 | 43.6 |
| Share Down | 19.0M | 52.1 | 77.3 | **85.2** | **71.5** | 1.0 | **44.5** |
| Share Up | 19.0M | 50.1 | 76.6 | 84.2 | 70.3 | 1.0 | 43.1 |
| Share Up & Down | **9.5M** | 50.8 | 77.1 | 84.3 | 70.7 | 1.0 | 43.4 |

modality encoders. We can see that: (1) Query-residual Adaption leads to a noticeable improvement, suggesting that maintaining query information is vital for cross-modal transfer learning. (2) PFA brings certain performance improvements without introducing extra parameters. (3) The weight-sharing scheme leads to competitive performance but with significantly fewer tunable parameters. (4) With these design considerations, our UniAdapter largely outperforms the standard Adapter, and reaches comparable or even better performance compared with Full fine-tuning for both retrieval tasks and reasoning tasks. This clearly shows the effectiveness and efficiency of our UniAdapter.

**Different Wight-sharing Strategies for UniAdapter.** Our UniAdapter shares a unified down-projection layer to enhance the cross-modal interaction and meanwhile reduce the tunable parameters. In Table 3, we make performance evaluation over our UniAdapter with different weight-sharing strategies. We observe that all sharing strategies achieve comparable performance but with 50%-70% fewer parameters compared with non-sharing (w/o Share), which demonstrates the effectiveness of the knowledge-sharing scheme. Among them, Share Down outperforms all the other strategies, indicating that modality-specific Up-projection layers are essential for cross-modal transfer learning. Considering both effectiveness and efficiency, we thus deploy Share Down as our weight-sharing strategy. We also provide a detailed analysis of the weight-sharing design in Appendix B.

**Comparisons with other parameter-efficient transfer learning methods.** We evaluated other parameter-efficient approaches in Table 4. We used the same hidden state $r = 512$ for all methods. For ST Adapter, which was designed for the visual encoder, we applied Parallel Adapter to the textual

Table 4: Comparisons between UniAdapter and other parameter-efficient transfer learning methods.

| Method | #Tunable | R@1 | R@5 | R@10 | MedR | R@Mean |
|---|---|---|---|---|---|---|
| LoRA (Hu et al., 2022a) | 56.6M | 43.3 | 70.6 | 79.7 | 2.0 | 64.5 |
| Adapter (Houlsby et al., 2019) | 28.4M | 49.6 | 75.9 | 82.8 | 2.0 | 69.4 |
| ST Adapter (Pan et al., 2022) | 42.5M | 49.9 | 75.6 | 83.2 | 2.0 | 69.6 |
| Parallel Adapter (He et al., 2022) | 28.4M | 49.9 | 76.2 | 83.0 | 2.0 | 69.7 |
| MAM Adapter (He et al., 2022) | 33.0M | 50.2 | 76.6 | 83.2 | 2.0 | 70.0 |
| UniAdapter (r=512) | 19.0M | 52.1 | 77.3 | 85.2 | 1.0 | 71.6 |

encoder as well as the multimodal encoder. Our UniAdapter achieved the best performance with the least number of tunable parameters, as can be observed from the results.

Table 5: Comparison to state-of-the-arts for video-text retrieval on MSR-VTT and Didemo. Input: the number×shape of inference frames. # Pretrain: the number of pre-training video/image-text pairs. The second best result is marked by underline.

| Method | Input | # Tunable | MSR-VTT | | | | Didemo | | | |
|---|---|---|---|---|---|---|---|---|---|---|
| | | | R@1 | R@5 | R@10 | MdR | R@1 | R@5 | R@10 | MdR |
| **Full fine-tuning:** | | | | | | | | | | |
| VIOLET (Fu et al., 2021) | 5×224 | 306M | 34.5 | 63.0 | 73.4 | - | 32.6 | 62.8 | 74.7 | - |
| All-in-one (Wang et al., 2022a) | 9×224 | 110M | 37.9 | 68.1 | 77.1 | - | 32.7 | 61.4 | 73.5 | 3.0 |
| CLIP-Hhiker (Bain et al., 2022) | 120×224 | 124M | 47.7 | 74.1 | 82.9 | - | - | - | - | - |
| OmniVL (Wang et al., 2022b) | 8×384 | 317M | 47.8 | **74.2** | **83.8** | - | 52.4 | **79.5** | 85.4 | - |
| LAVENDER (Li et al., 2023b) | 32×224 | 308M | 40.7 | 66.9 | 77.6 | - | 53.4 | 78.6 | 85.3 | - |
| SINGULARITY (Lei et al., 2023) | 16×224 | 209M | 41.5 | 68.7 | 77.0 | - | **53.9** | 79.4 | 86.9 | - |
| **Frozen backbone:** | | | | | | | | | | |
| CLIP-Prompt (Ju et al., 2022) | 16×224 | 6.4M | 36.7 | 64.6 | - | - | - | - | - | - |
| UniAdapter (ours, r=128) | 8×224 | **4.8M** | 49.7 | 71.9 | 81.5 | 2.0 | 49.0 | 75.5 | 83.3 | 2.0 |
| UniAdapter (ours, r=512) | 8×224 | 19.0M | 50.6 | 73.4 | 81.6 | **1.0** | 52.1 | 77.3 | 85.2 | **1.0** |
| UniAdapter (ours, r=512) | 16×224 | 19.0M | **50.5** | 73.9 | 81.7 | **1.0** | 53.7 | 78.3 | **87.2** | **1.0** |

Table 6: Comparison to the state-of-the-arts for the VideoQA task. # Tunable: the number of tunable parameters.

| Method | # Tunable | MSRVTT-QA Test Acc |
|---|---|---|
| **Full fine-tuning:** | | |
| MERLOT (Zellers et al., 2021) | 233M | 43.1 |
| All-in-one (Wang et al., 2022a) | 110M | 44.3 |
| SINGULARITY (Lei et al., 2023) | 209M | 43.5 |
| VIOLETv2 (Fu et al., 2023) | 308M | 44.5 |
| VINDLU (Cheng et al., 2023) | 201M | 44.6 |
| LAVENDER (Li et al., 2023b) | 308M | **45.0** |
| **Frozen backbone:** | | |
| UniAdapter (ours, r=128) | **4.8M** | 44.2 |
| UniAdapter (ours, r=512) | 19.0M | 44.7 |

Table 7: Comparison to the state-of-the-arts for the VQA task. # Tunable: the number of tunable parameters. *our implementation.

| Method | # Tunable | VQA | |
|---|---|---|---|
| | | test-dev | test-std |
| **Fine-tuning with huge backbone:** | | | |
| Flamingo (Alayrac et al., 2022) | 10.6B | 82.1 | 82.0 |
| BLIP-2 (Li et al., 2023a) | 1.2B | 82.3 | 82.2 |
| **Full fine-tuning:** | | | |
| OSCAR (Li et al., 2020) | 330M | 73.61 | 73.82 |
| ALBEF (Li et al., 2021) | 266M | 75.84 | 76.04 |
| BLIP* (Li et al., 2022) | 337M | **77.44** | **77.48** |
| **Frozen backbone:** | | | |
| UniAdapter (ours, r=128) | **4.8M** | 73.72 | 73.71 |
| UniAdapter (ours, r=512) | 19.0M | 75.44 | 75.56 |

## 4.3 COMPARISON TO THE STATE-OF-THE-ARTS

In this subsection, we compare our proposed UniAdapter to the recent state-of-the-art methods on a wide range of vision-language downstream tasks. Below we briefly introduce each downstream task and the corresponding tuning strategy (see more details in Appendix Section 4).

**Video-text Retrieval.** We first evaluate UniAdapter for video-text retrieval on MSR-VTT and Didemo in Table 5. We froze the pre-trained backbone, and fine-tune the tunable parameters (UniAdapter) following BLIP (Li et al., 2022). It can be observed that UniAdapter (4.8M) significantly outperforms the latest parameter-efficient method CLIP-Prompt (Ju et al., 2022) (49.7 vs. 36.7 for Text-to-Video R@1 on the MSR-VTT 1k-A test set) but with fewer tunable parameters (4.8M vs. 6.4M) and less input (8×224 vs. 16×224). UniAdapter even outperforms those full fine-tuning methods specially designed for video-text retrieval on both MSR-VTT and Didemo with significantly fewer tunable parameters. When trained with more tunable parameters (19.0M, still largely fewer than full fine-tuning), UniAdapter could further boost the performance.

**Video Question Answering.** To further export the potentiality of our UniAdapter, we evaluate it for the VideoQA task on the MSRVTT-QA dataset. Different from retrieval tasks, VideoQA requires the model to predict an answer given a video and a question. We follow (Li et al., 2022) to utilize an additional cross-modal decoder to generate answers. To reduce the tunable parameters, we share UniAdapter for both the multimodal encoder and multimodal decoder (see more details in Appendix Section 3). As shown in Table 6, even without utilizing large-scale video datasets devoted to the VideoQA task and fine-tuning on full parameters, our UniAdapter

Table 8: Comparison on the caption task. # Tunable: the number of tunable parameters.

| Method | # Tunable | COCO Caption | |
|---|---|---|---|
| | | CIDER | B@4 |
| **Fine-tuning with huge backbone:** | | | |
| SimVLM (Wang et al., 2022d) | 1.4B | 40.6 | 143.3 |
| BlIP-2 (Li et al., 2023a) | 1.1B | 42.4 | 144.5 |
| Flamingo (Alayrac et al., 2022) | 10.6B | 42.4 | 138.1 |
| **Full fine-tuning:** | | | |
| VL-T5/BART (Cho et al., 2021) | 165M | - | 71.30 |
| OSCAR (Li et al., 2020) | 345M | 37.4 | 127.8 |
| VinVL (Chen et al., 2020) | 345M | 38.2 | 129.3 |
| BLIP* (Li et al., 2022) | 337M | 39.7 | 133.3 |
| **Frozen backbone:** | | | |
| UniAdapter (ours, r=512) | 19.0M | **40.0** | **133.5** |

Table 9: Comparison to the state-of-the-arts for image-text retrieval on MSCOCO (5K) and Flickr30K. # Tunable: the number of tunable parameters. The second best result is marked by underline.

| Method | # Tunable | MSCOCO TR | | | MSCOCO IR | | | Flcikr TR | | | Flcikr IR | | |
|---|---|---|---|---|---|---|---|---|---|---|---|---|---|
| | | R@1 | R@5 | R@10 | R@1 | R@5 | R@10 | R@1 | R@5 | R@10 | R@1 | R@5 | R@10 |
| **Fine-tuning with huge backbone:** | | | | | | | | | | | | | |
| BEIT-3 (Wang et al., 2023) | 1.9B | 84.8 | 96.5 | 98.3 | 67.2 | 87.7 | 92.8 | 98.0 | 100.0 | 100.0 | 90.3 | 98.7 | 99.5 |
| BLIP-2 (Li et al., 2023a) | 1.2B | 85.4 | 97.0 | 98.5 | 68.3 | 87.7 | 92.6 | - | - | - | - | - | - |
| **Full fine-tuning:** | | | | | | | | | | | | | |
| UNITER (Chen et al., 2020) | 330M | 65.7 | 88.6 | 93.8 | 52.9 | 79.9 | 88.0 | 87.3 | 98.0 | 99.2 | 75.6 | 94.1 | 96.8 |
| OSCAR (Li et al., 2020) | 330M | 73.5 | 92.2 | 96.0 | 57.5 | 82.8 | 89.8 | - | - | - | - | - | - |
| ALBEF (Li et al., 2021) | 210M | 77.6 | 94.3 | 97.2 | 60.7 | 84.3 | 90.5 | 95.9 | 99.8 | **100.0** | 85.6 | 97.5 | **98.9** |
| BLIP (Li et al., 2022) | 223M | **81.9** | **95.4** | 97.8 | **64.3** | **85.7** | **91.5** | **97.3** | 99.9 | **100.0** | 87.3 | 97.6 | **98.9** |
| **Frozen backbone:** | | | | | | | | | | | | | |
| UniAdapter (ours, r=128) | **4.8M** | 79.8 | 94.2 | 97.5 | 62.3 | 84.5 | 90.8 | 97.1 | **100.0** | **100.0** | 86.5 | 97.4 | 98.8 |
| UniAdapter (ours, r=512) | 19.0M | 80.1 | 94.6 | 97.4 | 62.6 | 84.6 | 90.9 | 97.1 | 99.9 | **100.0** | 86.4 | 97.4 | **98.9** |

outperforms most full fine-tuning competitors. These results suggest that parameter-efficient transfer learning may be a more effective paradigm for the VideoQA task.

**Image-text Modeling.** We also evaluate UniAdapter for image-language domains, including visual language reasoning tasks (VQAv2, in Table 7), and image-text retrieval tasks (MSCOCO and Flickr30K, in Table 9) and image caption task (MSCOCO, in Table 8).

For VQA in Table 7 and Caption in Table 8, fine-tuned on only 5% parameters, our UniAdapter can achieve competitive or even better performance compared with fully fine-tuned methods. For image-text retrieval in Table 9, our UniAdapter performs comparably on MSCOCO and almost equally on Flickr30K but with only 2–10% tunable parameters.

## 4.4 TRAINING EFFICIENCY AND STORAGE COST

The performance of parameter-efficient transfer learning methods is typically sensitive to the number of tunable parameters. We thus conduct experiments with different values of bottleneck dimension $r \in \{64, 128, 256, 512, 768\}$ in Figure 3 (a). We can observe that our UniAdapter is effective on a wide range of bottleneck dimension $r$ and achieves a slight performance improvement (or maintains the accuracy stably) when $r$ scales up. This suggests that our UniAdapter is not sensitive to the bottleneck dimension $r$ and we could select $r$ according to the practical requirements.

Compared with the standard Adapter, our UniAdapter can achieve higher performance (43.6 vs. 43.2) but with 5.9× fewer tunable parameters. When utilizing the same parameters (especially on small ones), our UniAdapter leads to further gains over the standard Adapter.

We compare training efficiency in Figure 3 (b). UniAdapter and full fine-tuning perform very comparably at the early stage. Then the full fine-tuning strategy drops quickly which may be due to the overfitting, while UniAdapter further boosts the performance. Meanwhile, UniAdapter is significantly faster than the standard Adapter (same performance with nearly 3× fewer training steps). We also report the relative training GPU hours and GPU memory cost for both retrieval and VQA tasks in Table 10, where the time (or memory) of full fine-tuning is taken as one unit.

Table 10: Comparison on the training time and GPU memory. * means that more resources required is mainly due to the additional momentum encoder applied by BLIP (Li et al., 2022) for retrieval task.

| | # Param | VQA | | Retrieval* | |
|---|---|---|---|---|---|
| | | Time | Mem | Time | Mem |
| Full fine-tuning | 100% | 1.00 | 1.00 | 1.00 | 1.00 |
| UniAdapter (r=128) | 1.4%-2.2% | 0.44 | 0.60 | 0.81 | 0.73 |
| UniAdapter (r=512) | 5.6%-8.5% | 0.47 | 0.61 | 0.86 | 0.76 |

## 5 CONCLUSION

In this paper, we propose UniAdapter for parameter-efficient cross-modal adaptation. UniAdapter unifies adapters for different modalities and their interactions with a knowledge-sharing design. By incorporating a small number of tunable parameters, we capitalize a frozen vision-language model to adapt to unified modalities (*e.g.*, image and video) as well as unified cross-modal downstream tasks (*e.g.*, retrieval and reasoning). Extensive evaluations on six cross-modal downstream benchmarks show that UniAdapter typically outperforms previous arts and even surpasses the full fine-tuning strategy. We believe this work will inspire further research on efficient cross-modal modeling tasks.

ACKNOWLEDGEMENTS

This work was supported by National Natural Science Foundation of China (62376274).

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

Table 11: Applying UniAdapter on different backbones, evaluated on Flickr image-text retrieval task.

| Method | Flcikr TR | | | Flcikr IR | | |
|---|---|---|---|---|---|---|
| | R@1 | R@5 | R@10 | R@1 | R@5 | R@10 |
| ALBEF (Full finetuning) | 95.9 | 99.8 | 100.0 | 85.6 | 97.5 | 98.9 |
| UniAdapter (Frozen backbone) | 95.6 | 99.9 | 100.0 | 84.6 | 97.3 | 98.6 |
| BLIP-Large (Full finetuning) | 97.4 | 99.8 | 99.9 | 87.6 | 97.7 | 99.0 |
| UniAdapter (Frozen backbone) | 96.9 | 99.9 | 100.0 | 87.5 | 97.9 | 99.1 |
| BEIT 3-Large (Full finetuning) | 97.1 | 100.0 | 100.0 | 87.5 | 97.9 | 99.1 |
| UniAdapter (Frozen backbone) | 96.3 | 99.9 | 100.0 | 87.6 | 97.9 | 99.2 |

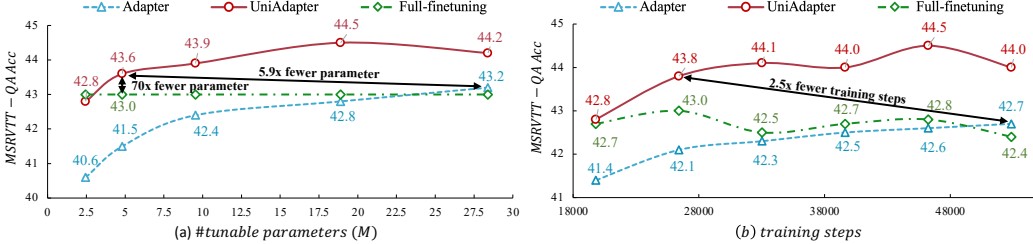

Figure 3: (a) Parameter efficiency comparison with standard Adapter and full fine-tuning. (b) Training efficiency comparison. We adopt the bottleneck dimension $r = 512$ for Adapter / UniAdapter.

# A   GENERALIZATION OF UNIADAPTER

We also experimented with the application of our UniAdapter to various backbone architectures, and the corresponding results are presented in Table 11. It is evident that our model achieves performance levels that are either comparable or even superior to full fine-tuning on ALBEF. Furthermore, we extended our investigation by applying our UniAdapter to larger backbones (BLIP-Large and BEIT 3-Large). Our UniAdapter consistently demonstrates either comparable or superior performance when compared to full fine-tuning. This observation underscores the potential advantages of our approach, particularly for larger models.

# B   DETAIL ANALYSIS FOR WEIGHT-SHARING STRATEGY

**Comparisons for different layers of the modality sharing.** We investigate the sharing layer for modality sharing in Table 12. Our findings indicate that sharing only 9-12 layers leads to a slight performance degradation, as shown in line 2 compared to line 1. However, when additionally sharing layers 5-8, it achieves higher performance with fewer tunable parameters than the non-sharing results. This sug-

Table 12: Comparisons for different layers of the modality sharing. # Tunable: the number of tunable parameters. The bottleneck dimension $r$ is set to $r = 512$ for Adapter.

| Layer | Layer | Layer | ShareAll | ShareDown | ShareUp |
|---|---|---|---|---|---|
| 1-4 | 5-8 | 9-12 | R@Mean | R@Mean | R@Mean |
| | | ✓ | 70.2 | 71.1 | 70.0 |
| | ✓ | ✓ | 70.5 | 71.6 | 70.2 |
| ✓ | | ✓ | 70.1 | 70.9 | 70.0 |
| ✓ | ✓ | ✓ | 70.7 | 71.5 | 70.3 |

gests that middle-layer modality-sharing is crucial for optimal performance. These findings are also consistent with the layer location for injecting UniAdapter, as shown in Table 15. Direct sharing for each layer could achieve optimal performance with the least number of parameters, as shown by the comparison between line 2 and line 4.

**Weight-sharing for Query-residual Adaption.** In cross-modal adaption (see Sec.3.3 of the main paper), the Query-residual Adaption shares weights with the textual adapter to avoid additional parameters. We compare it with utilizing an additional adapter as Query-residual adapter in Table 13.

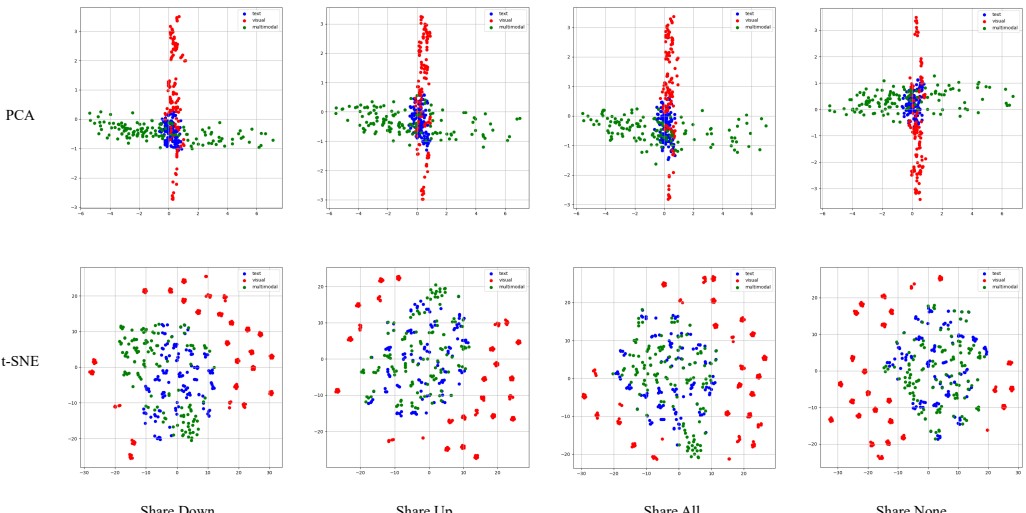

Figure 4: t-SNE and PCA visualizations of input representations of UniAdapter at the 12th layer from different modality models. Experimental data is derived from 128 randomly chosen images from Flickr30K. The inputs to the UniAdapter from various modalities exhibit similar spatial distributions, suggesting that they reside within the same space (or sharing several spaces across each distribution).

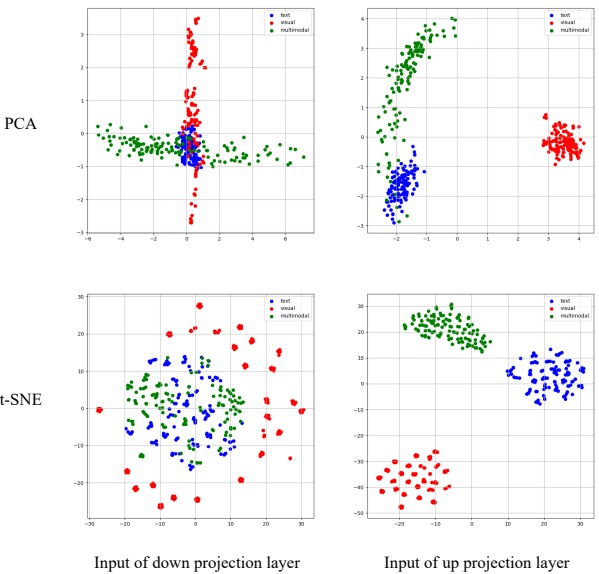

Figure 5: t-SNE and PCA visualizations of input representations of UniAdapter at the 12th layer from different modality models. Experimental data is derived from 128 randomly chosen images from Flickr30K. The inputs to the down projection layer of UniAdapter from various sharing-strategy exhibit similar spatial distributions. This finding suggests that these representations reside within a shared space or have overlapped distributions across different modalities. Different modalities' data distributions occupy distinct spaces when it comes to the up-projection layer. This finding validates that sharing the up-projection layer negatively impact the model's performance, as it appears to interfere with the distinct representation spaces required for different modalities.

We find that sharing weight with the textual adapter leads to better performance but with fewer tunable parameters.

Query9354: A red truck is burning while three men talk about a car.

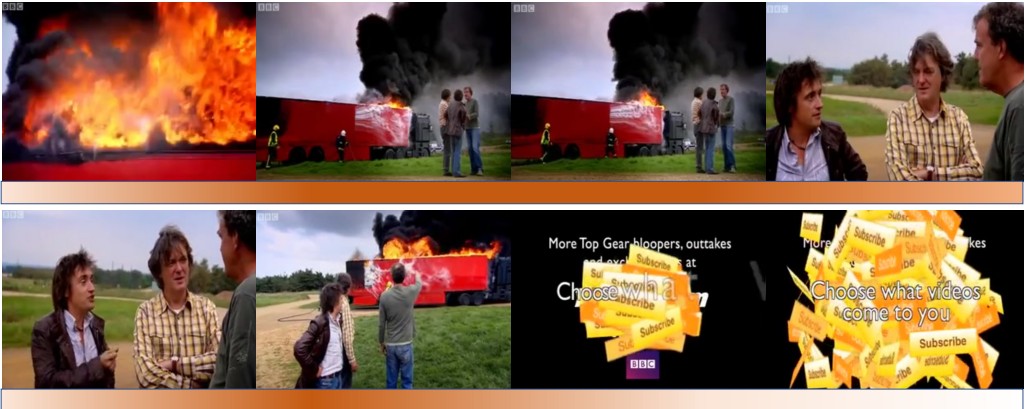

Query8069: An astronaut is looking at a flag.

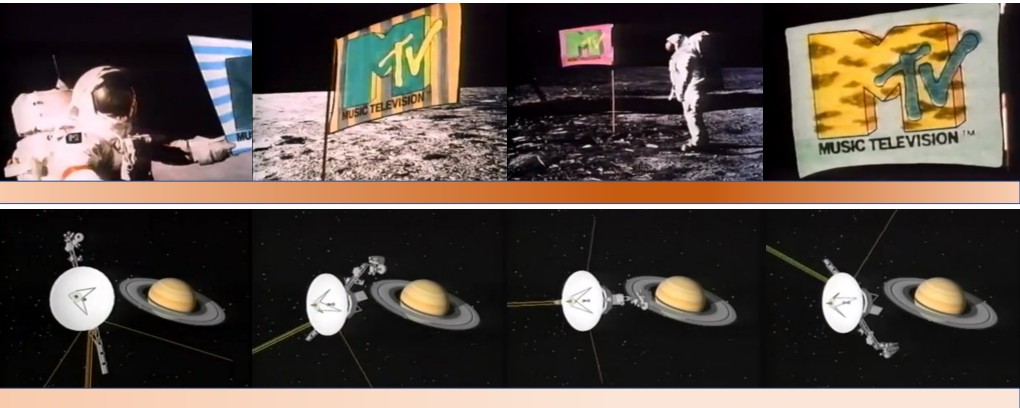

Figure 6: Visualization results for PFA mechanism (color bar below represents the weights assigned to various tokens). Tokens that exhibit a close relationship with the text query are assigned higher weights. On the other hand, noise tokens, which bear little relevance or contribute insignificantly to the text query, are assigned lower weights, effectively reducing their influence.

Table 13: Weight-sharing for Query-residual Adaption on Didemo. # Tunable: the number of tunable parameters. The bottleneck dimension $r$ is set to $r = 512$ for Adapter.

| Method | # Tunable | R@1 | R@5 | R@10 | R@Mean | MedR |
|--------|-----------|-----|-----|------|--------|------|
| Adapter | **28.4M** | 49.9 | 76.2 | 83.0 | 69.7 | 2.0 |
| +Q w/o share | 37.8M | 50.9 | 77.1 | 83.3 | 70.4 | **1.0** |
| +Q w/ share | **28.4M** | **51.1** | **77.2** | **84.1** | **70.8** | **1.0** |

We conduct further visualizations on the 12th layer inputs from different modality modules in our UniAdapter in Figure 4. Using joint t-SNE and PCA, we found: Inputs representations to the UniAdapter from various modalities exhibit similar spatial distributions, suggesting that they reside within a shared space (or overlapped spaces across distributions). This supports the effectiveness of our down-projection sharing strategies.

Table 14: Query-residual mechanism for UniAdapter on cross-modal modality. # Tunable: the number of tunable parameters. The bottleneck dimension $r$ is set to $r = 512$ for UniAdapter.

| Method | # Tunable | R@1 | R@5 | R@10 | R@Mean | MedR |
|--------|-----------|-----|-----|------|--------|------|
| UniAdapter | 19.0M | 51.6 | 76.5 | 83.6 | 70.6 | 1.0 |
| +Q | 19.0M | **52.1** | **77.3** | **85.2** | **71.5** | 1.0 |

We also follow this effective and efficient strategy to design UniAdapter, where the cross-modal up-projection branch utilizes a Query-residual up-projection layer (shared knowledge with textual up-projection layer as shown in Figure 2 (b) in the main paper). We can see from Table 14 that this strategy leads to better performance without introducing extra parameters.

## C  THE GLOBAL POSITION FOR INJECTING UNIADAPTER.

We investigated the effects of adding adapters to certain layers in our approach as below (Table B), and the results reveal two key findings.

First, we found that adding an adapter to layers 5-8 achieved the highest performance, which is interesting because traditional single modality parameter-efficient approaches typically perform best when an adapter is added to higher layers. This result may suggest that cross-modal fusion occurs more frequently in the middle layers rather than the higher layers. Second, we found that adding adapters to layers 5-12 achieved comparable performance to adding them to all layers, indicating that the earlier layers tend to learn more general features that are task-irrelevant. This finding has important implications for the design of parameter-efficient transfer learning models, as it suggests that adding adapters to all layers may not always be necessary for achieving optimal performance.

Table 15: The global position for injecting UniAdapter. # Tunable: the number of tunable parameters. The bottleneck dimension $r$ is set to $r = 512$ for Adapter.

| Layer Position | | | #Tunable | Didemo | | | | |
|---|---|---|---|---|---|---|---|---|
| 1-4 | 5-8 | 9-12 | | R@1 | R@5 | R@10 | MedR | R@Mean |
| ✓ | | | 6.3M | 45.8 | 74.3 | 82.6 | 2.0 | 67.5 |
| | ✓ | | 6.3M | 51.1 | 76.5 | 83.5 | 1.0 | 70.4 |
| | | ✓ | 6.3M | 48.8 | 74.9 | 82.2 | 1.0 | 68.6 |
| | ✓ | ✓ | 12.7M | 52.0 | 77.4 | 84.0 | 1.0 | 71.2 |
| ✓ | ✓ | ✓ | 19.0M | 52.1 | 77.3 | 85.2 | 1.0 | 71.6 |

## D  UNIADAPTER FOR VQA TASKS

Hybrid-stream vision-language models (Li et al., 2021; Bain et al., 2022) consider visual question answering as an answer generation task. As shown in Figure 7, an image/video-question is first encoded into multimodal embeddings and then inputted into an additional multimodal decoder during fine-tuning. However, the additional answer decoder leads to heavier tunable parameters. Instead of introducing additional UniAdapter specially designed for the multimodal decoder, we choose to directly share the weight with the UniAdapter adopted in the multimodal encoder. This choice is simple yet effective, and can avoid the additional parameter cost.

## E  DETAILS OF DOWNSTREAM TASKS

### E.1  HYPERPARAMETER SETTING

We list hyperparameters for downstream tasks in Table 16. For image-text downstream tasks, we directly use hyperparameters applied in BLIP (Li et al., 2022) without modifying. For video-text downstream task, we follow previous works (Lei et al., 2021; Ju et al., 2022) to uniformly sample $N = 8$ frames for training and $N = 16$ frames for inference.

### E.2  VIDEO-TEXT RETRIEVAL

**MSR-VTT** (Xu et al., 2016) is a popular video-text retrieval dataset. We follow recent works (Lei et al., 2021; Luo et al., 2021) to adopt the 1k-A split (with 9,000/1,000 videos) for training/testing. **DiDeMo** (Hendricks et al., 2017) consists of 10K videos and 40K sentences. Each sentence includes the temporal localization information. Following Frozen in Time (Bain et al., 2021), we conduct the paragraph-to-video retrieval task, where all descriptions in the same video are concatenated into a single description (*i.e.*, one paragraph).

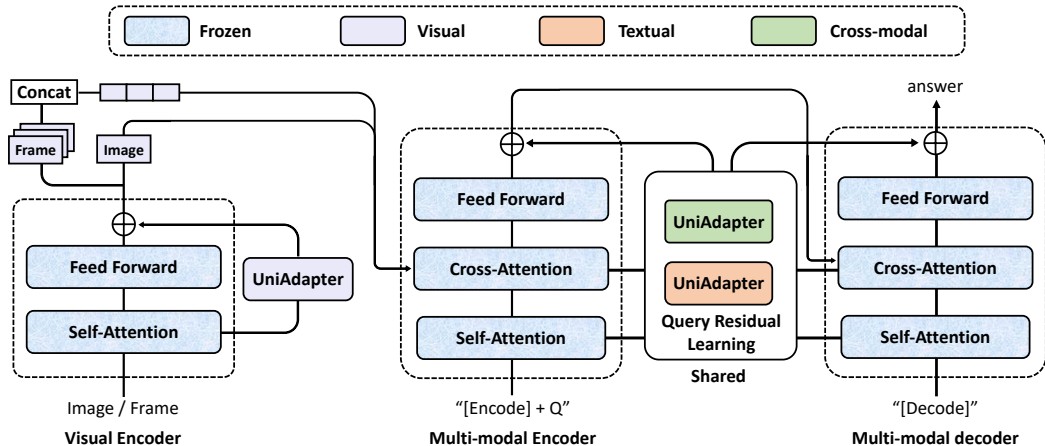

Figure 7: A semantic illustration of the proposed UniAdapter for the VQA tasks. We adopt a single UniAdapter for the multimodal encoder as well as multimodal decoder.

Table 16: Parameter-efficient fine-tuning hyperparameters for each task.

| Config | Video-text Retrieval | | Image-text Retrieval | | VQA | |
| | MSR-VTT | Dideomo | MSCOCO | Flickr30K | MSRVTT-QA | VQAv2 |
| --- | --- | --- | --- | --- | --- | --- |
| optimizer | AdamW | AdamW | AdamW | AdamW | AdamW | AdamW |
| learning rate | 1e-5 | 1e-5 | 1e-5 | 1e-5 | 2e-5 | 2e-5 |
| schedule | cosine decay | cosine decay | cosine decay | cosine decay | cosine decay | cosine decay |
| batch size | 64 | 64 | 256 | 256 | 24 | 128 |
| epochs | 5 | 10 | 5 | 6 | 10 | 10 |
| training input | 8x224 | 8x224 | 384 | 384 | 8x384 | 480 |
| inference input | 16x224 | 16x224 | 384 | 384 | 16x384 | 480 |

### E.3 IMAGE-TEXT RETRIEVAL

**MSCOCO** (Lin et al., 2014) is a large image-text dataset of 123,287 images, where each image is annotated with 5 captions. As in (Kim et al., 2021), we adopt the Karpathy split of MSCOCO: 5,000 images for testing, another 5,000 for validation, and the rest 113,287 images for training. **Flickr30K** (Plummer et al., 2015) contains 31,000 images and 158,915 captions totally. Each image is often annotated with 5 captions. Following the split in (Frome et al., 2013), we use 1,000 images for testing, another 1,000 for validation, and the rest for training.

### E.4 VISUAL QUESTION ANSWERING

**MSRVTT-QA** (Xu et al., 2017) is a popular video-question answering dataset. We employ the standard split as in ClipBERT (Lei et al., 2021), which contains 244K open-ended questions on 10K MSRVTT videos. **VQAv2** (Goyal et al., 2017) is a visual question answering dataset constructed from COCO, which has 83k/41k/81k images for training/validation/testing. Following previous works (Li et al., 2021; 2022), we utilize both training and validation sets of VQAv2, question-answer pairs from Visual Genome (Krishna et al., 2017) for training. We report the results on the test-dev and test-std splits.

### E.5 CAPTION

We utilize COCO's Karpathy split following previous work (Li et al., 2021). During inference, we use beam search (beam size of 3), and the maximum generation length of 20.

## F EVALUATION METRICS

We adopt two widely-used metrics in cross-modal retrieval: Recall at K (R@K, K= $1, 5, 10$), and Median Rank (MdR). R@K means the percentage of correct matching in the K nearest points, and MdR measures the median rank of target items in the retrieved ranking list. We also report

Query8110: A computer generated cartoon figure operates a control panel while another character sleeps in the background.

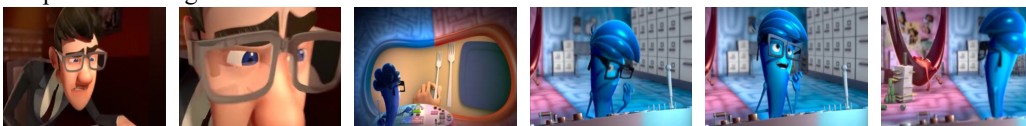

Query7468: A man jumps onto a ledge of a building.

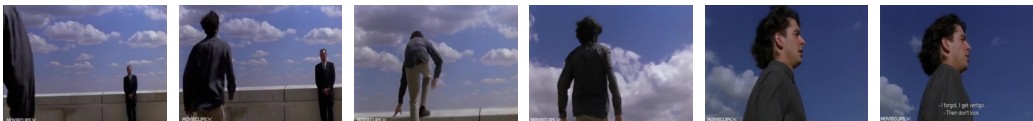

Query8069: An astronaut is looking at a flag.

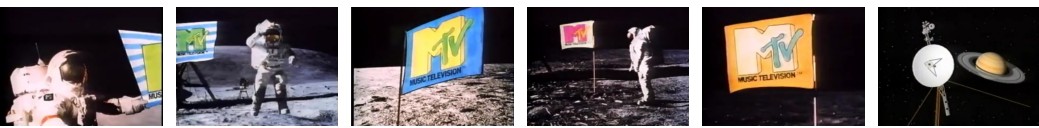

Query9354: A red truck is burning while three men talk about a car.

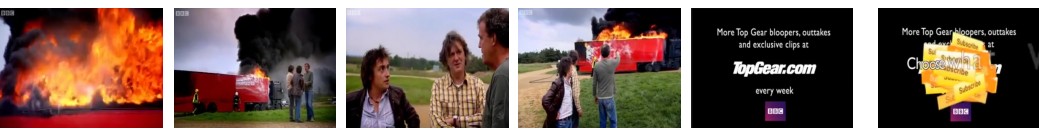

Query7793: Flight is shaken and the pilots trying to land the flight while they opened the air.

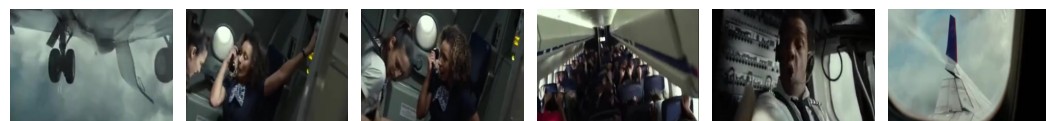

Query9779: Fireworks are being lit and exploding in a night sky.

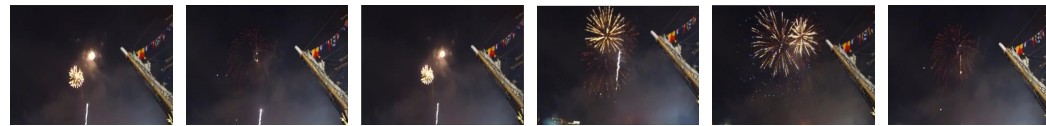

Figure 8: Text-to-video retrieval examples on the MSR-VTT test set.

additional metrics named 'R@Mean' in our ablation study, which averages all recall metrics for overall evaluation. Following previous works (Li et al., 2020; Lei et al., 2021; Li et al., 2021), we also report accuracy (Acc) in visual question answering task.

## G  VISUALIZATION RESULTS

In Figures 8–9, we show the visualization results for the text-to-video retrieval task and video question answering task, respectively. Even tuning on small parameters, our UniAdapter shows strong semantic understanding/reasoning ability.

## H  LIMITATION AND BROADER IMPACTS

In this paper, UniAdapter focuses on retrieval and VQA tasks in the multimodal domain. In the future, we hope to use our UniAdapter for more complex tasks such as cross-modal generation. Parameter-efficient transfer learning enables many powerful models to be used by the general public. Therefore, it is crucial to conduct a comprehensive analysis of the potential consequences and adopt responsible practices to address any negative impacts.

Query 169863: What is a table tennis match between two chinese players and the winner doing?
Target Answer: celebrate
UniAdapter Answer: celebrate

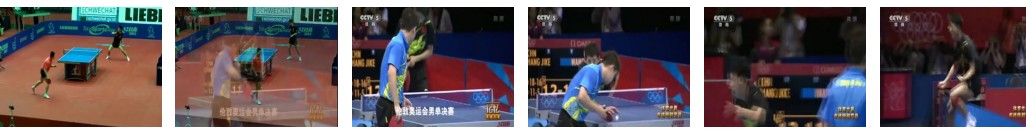

Query 168667 : Who is a woman fixing?
Target Answer: stroller
UniAdapter Answer: stroller

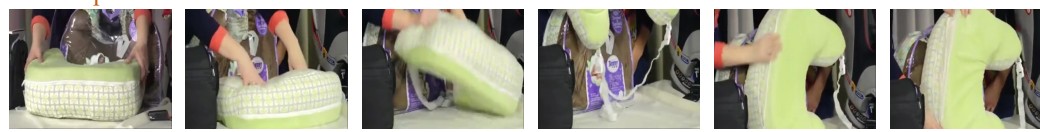

Query 168815: What is two car crashed in doing?
Target Answer: race
UniAdapter Answer: race

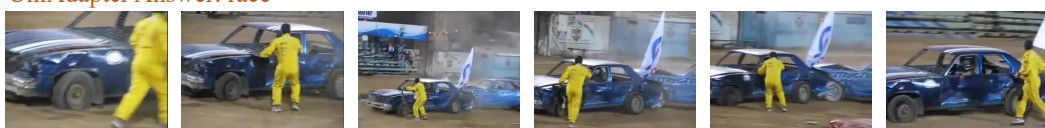

Query 168971: How many cars are shown in a video game while music plays in the background?
Target Answer: three
UniAdapter Answer: three

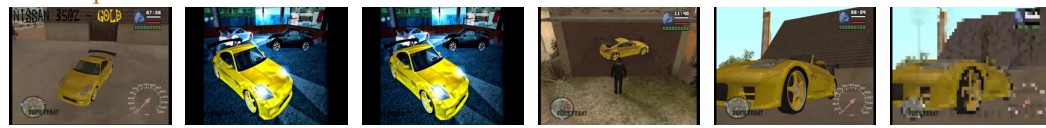

Query 169627: What is a woman adds green vegetables to a tiny pot of doing?
Target Answer: boil
UniAdapter Answer: boil

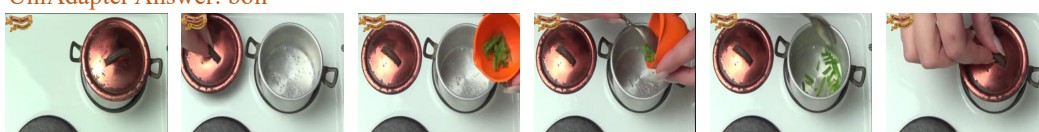

Figure 9: Video question answering examples on the MSRVTT-QA test set.

