# OpenReview forum: "UniAdapter: Unified Parameter-Efficient Transfer Learning for Cross-modal Modeling"
_ICLR.cc/2024/Conference — ICLR 2024 poster_

### Official Review · Reviewer_2kKp · 2023-10-16

**Soundness:** 3 good
**Presentation:** 3 good
**Contribution:** 3 good
**Rating:** 8
**Confidence:** 4

**Summary:**

This paper proposed a parameter-efficient adapter for fine-tuning vision-language foundation models. The unified architecture for the proposed adapter can not only support visual or textual single modality, but also support both together by sharing knowledge together in cross-modality. The contributions for this adapter are threefold, 1) residual learning for language queries within adapter and in multi-modal encoder; 2) knowledge sharing of cross-modality only in down-projection layers; and 3) a parameter-free frame-aware attention mechanism to extend image approach to video inputs. Six cross-modal experiments are validated on the proposed adapter, and the authors demonstrate it outperforms other SOTA methods on both accuracy and tunable parameters. Furthermore, it can achieve or even surpass full-fine tuning results on these datasets.

The authors have addressed all my comments carefully during rebuttal phase. I have no other concerns and it is indeed a stronger paper now. Hence I raise my rating.

**Strengths:**

1. Good motivation for the topic and its approach. Efficient fine-tuning is indeed needed for VLM so that transfer learning is feasible for wide range of applications. This makes both academia and individual works on fine-tuning large models possible who usually have insufficient hardware resources.
2. Clear and nice writing. It is easy to understand its concept and contributions. Appreciate it.
3. Novelty on residual learning for textual information. Good observation and approach to improve the performance of UniAdapter. The residual part for both inside adapter and in multi-modal encoder parts are great ideas to apply residual learning for text.
4. Strong experiments. Extensive experiments and very detailed ablation studies to support its argument.
5. Great contributions for releasing the code.

**Weaknesses:**

1. Although the authors propose using residual learning to preserve the integrity of language queries during the cross-attention process in multimodal encoders, it is not clear why only textual queries need to be preserved, not visual queries. Specifically, in Fig. 2(b), why if text info may be missing which needs a residual learning, why not visual info after up-projection linear layer not be added to the cross-modal output? Similarly for Fig. 2(a), why there is no extra UniAdapter for visual modal be needed i mluti-modal encoder? This needs to be explained carefully, with evidence/experiments.
2. It is not clear what is "the noise and misalignment problem in videos" and how the proposed PFA can mitigate these issues. Need more insights be explained or visualizations, not only demonstrated by ablations.
3. Regarding equation (7), how to justify only using text token feature is good in this case? How about other features f^{t}_{CLS,i}?
4. In Table 2, need ablation studies on top of UniAdapter, not Adapter. On Adapter is good and helpful, but it is needed to be put on top of the proposed solution to see its full benefit, i.e., +Query-residual, +PFA, +Weight-sharing all in Full UniAdapter. Would like to see the enhancement on the full version.
5. There is supposed to have two textual query-residual learnings need to be validated, i.e., in both Fig. 2(a) and Fig. 2(b). However, in Table 2 there is only one +Query-residual Adaption. Is this a combined experiment for both residual learning? Would like to see this ablation with separate results.

**Questions:**

See weaknesses. Please respond to all the questions and request there.

**Details Of Ethics Concerns:**

None.

---

> ### Author Response · Authors · 2023-11-18
> **Response to Reviewer 2kKp (Part 1/2)**
>
> Thank you for the positive comments and insightful suggestions. Your insightful questions and valuable suggestions have been immensely helpful in enhancing the paper's quality.
>
> **Q1: Although the authors propose using residual learning to preserve the integrity of language queries during the cross-attention process in multimodal encoders, it is not clear why only textual queries need to be preserved, not visual queries. Specifically, in Fig. 2(b), why if text info may be missing which needs a residual learning, why not visual info after up-projection linear layer not be added to the cross-modal output? Similarly for Fig. 2(a), why there is no extra UniAdapter for visual modal be needed in mluti-modal encoder? This needs to be explained carefully, with evidence/experiments.**
>
> **A:** Thank you for your thoughtful question regarding our decision to focus on preserving textual information over visual information in our UniAdapter model.
>
> * Our decision to focus on textual information preservation **instead of visual** is informed by the operational structure of prevalent multimodal cross-attention mechanisms, such as those in ALBEF, BLIP, and Perceiver architectures. In these models, the visual information (the final token sequence output $f_v$ from the visual encoder) is inherently integrated into the each layer of the cross-attention module. This is exemplified in the following equation from our paper:
>
>    $$ h_i = q_i + MCA(Q = q_i, K = f_v, V = f_v) $$
>
>     In this design, $f_v$ consistently serves as both the key and value in each layer of the cross-attention module, ensuring that visual information is intrinsically embedded at every layer.
>
> * Due to this structural design, visual information is naturally preserved and maintained throughout the cross-attention layers. This effective integration prevents the loss or diminishment of visual information as it progresses through these layers. Consequently, this inherent preservation of visual information within the cross-attention mechanism eliminates the need for additional measures, such as extra up-projection linear layers or UniAdapters, specifically for the visual modality in the multimodal encoder.
> * Our experimental trials also support this design choice. We tested the addition of extra mechanisms for visual information preservation and found that it did not lead to performance improvements. Instead, it introduced unnecessary parameter overhead.
>
> Based on these findings, we concluded that there was no necessity for additional mechanisms like extra UniAdapters for the visual modality or additional up-projection linear layers for the visual information in the cross-modal output of our UniAdapter model.
>
> **Q2: It is not clear what is "the noise and misalignment problem in videos" and how the proposed PFA can mitigate these issues. Need more insights be explained or visualizations, not only demonstrated by ablations.**
>
> **A:** The core functionality of our PFA is to enhance the relevance of tokens that align closely with the text query, while reducing the impact of noise tokens that are unrelated. This selective enhancement and suppression of tokens is particularly crucial in addressing the common challenge of semantic misalignment between video frames and text descriptions in video-language modeling. Misalignment often occurs when certain video frames do not correspond meaningfully to the given text descriptions, creating noise and ambiguity in the model's interpretation.
>
> To offer a more tangible understanding of how PFA operates, we have included specific visualizations in Figure 6 of our paper. These visualizations demonstrate the dynamic operation of PFA in action: tokens that exhibit a close relationship with the text query are assigned higher weights, thereby gaining increased prominence. On the other hand, noise tokens, which bear little relevance or contribute insignificantly to the text query, are assigned lower weights, effectively reducing their influence.

---

> ### Author Response · Authors · 2023-11-18
> **Response to Reviewer 2kKp (Part 2/2)**
>
> **Q3: Regarding equation (7), how to justify only using text token feature is good in this case? How about other features $f^{t}_{CLS,i}$?**
>
> **A:** Thank you for your insightful question regarding our decision to use the text token feature $f^t_{CLS}$. Our choice is primarily based on the effectiveness of the contrastive learning process, in which the overall semantic representation of a sentence is efficiently encapsulated into the $f^t_{CLS}$ token. This makes $f^t_{CLS}$ an optimal representation for the entire sentence in our model.
>
> Theoretically, incorporating a more granular calculation that includes each token and text token could potentially yield better results. However, such  approach would require the introduction of an additional module to handle the alignment of varying token lengths. To balance model efficiency and simplicity, we chose $f^t_{CLS}$ as our representative token.
>
>
>
> |  | $f^t_{CLS}$ | mean($f^t_{CLS,i}$) | max($f^t_{CLS,i}$) | $f^t_{CLS}$ + mean($f^t_{CLS,i}$) |
> | -------- | -------- | -------- |-------- |-------- |
> | Didemo R@1     |  52.1    | 51.9     | 51.2 | 52.0 |
>
>
> Additionally, we experimented with using the average or maximum of all $f^t_{CLS,i}$ tokens as an alternative. These variations, however, did not show any significant performance differences in our experiments, further validating our decision to use $f^t_{CLS}$ for maintaining both model efficiency and performance.
>
>
>
> **Q4: In Table 2, need ablation studies on top of UniAdapter, not Adapter. On Adapter is good and helpful, but it is needed to be put on top of the proposed solution to see its full benefit, i.e., +Query-residual, +PFA, +Weight-sharing all in Full UniAdapter. Would like to see the enhancement on the full version.**
>
> **A:** Thank you for your valuable feedback on the necessity of providing more explicit ablation studies on the full UniAdapter model in Table 2. We realized that this progression was not as clear as it could be in our original presentation. To address this, we have revised Table 2 to make it more clearly.
>
> **Q5: There is supposed to have two textual query-residual learnings need to be validated, i.e., in both Fig. 2(a) and Fig. 2(b). However, in Table 2 there is only one +Query-residual Adaption. Is this a combined experiment for both residual learning? Would like to see this ablation with separate results.**
>
> **A:** Thank you for your suggestion regarding the need for separate validation of the two textual query-residual learnings. We understand the importance of this clarification and have included detailed ablation studies in the appendix of our paper, specifically in Tables 14 and 15. We also show these below:
>
> **Weight-sharing for Query-residual Adaption:** In cross-modal adaption (see Sec.3.3 of the main paper), the Query-residual Adaption shares weights with the textual adapter to avoid additional parameters. We compare it with utilizing an additional adapter as Query-residual adapter in Table 14. We find that sharing weight with the textual adapter leads to better performance but with fewer tunable parameters.
> | Method         | Tunable | R@1  | R@5  | R@10 | R@Mean | MedR |
> |----------------|---------|------|------|------|--------|------|
> | Adapter        | 28.4M   | 49.9 | 76.2 | 83.0 | 69.7   | 2.0  |
> | +Q w/o share   | 37.8M   | 50.9 | 77.1 | 83.3 | 70.4   | 1.0  |
> | +Q w/ share    | 28.4M   | 51.1 | 77.2 | 84.1 | 70.8   | 1.0  |
>
> **UniAdapter's Query-Residual Up-Projection Layer:** We also follow this effective and efficient strategy to design UniAdapter, where the cross-modal up-projection branch utilizes a Query-residual up-projection layer (shared knowledge with textual up-projection layer as shown in Figure 2 (b) in the main paper). We can see from Table 15 that this strategy leads to better performance without introducing extra parameters.
>
> | Method     | Tunable | R@1  | R@5  | R@10 | R@Mean | MedR |
> |------------|---------|------|------|------|--------|------|
> | UniAdapter | 19.0M   | 51.6 | 76.5 | 83.6 | 70.6   | 1.0  |
> | +Q         | 19.0M   | 52.1 | 77.3 | 85.2 | 71.5   | 1.0  |
>
>
> Thanks again for your time and effort! For any other questions, please feel free to let us know during the rebuttal window.

---

> ### Author Response · Authors · 2023-11-19
>
> Thanks again for spending a huge amount of time on our paper, which has helped us improve the quality and clarity of the paper! We are glad to see that our response has addressed most of your concerns.
>
> Thanks for your time and efforts again!
>
> Best, \
> Authors

---

### Official Review · Reviewer_1KDb · 2023-10-29

**Soundness:** 3 good
**Presentation:** 3 good
**Contribution:** 2 fair
**Rating:** 3
**Confidence:** 4

**Summary:**

This paper introduces UniAdapter as a method that unifies unimodal and multimodal adapters for efficient cross-modal adaptation in vision-language models. The key components of UniAdapter are:
- Knowledge sharing design: use a shared down-projection layer among all adapters, while learning modality-specific up-projection layers
- Additional residual connection for language queries
- Parameter-free frame-aware attention to bring together video and image modalities. This is achieved by emphasizing tokens within important frames while suppressing those in noisy or irrelevant frames during cross-attention.

The proposed method emphasizes reductions in the number of tunable parameters while achieving competitive practical performance.

**Strengths:**

1. Quality & Significance: The problem of using a unified adapter architecture (and potentially shared weights) for modeling single-modal and multi-modal interactions is interesting. While the algorithm design lacks originality, the empirical evaluation (e.g., the ablations studies) is good, and a wide array of baselines is considered.
2. Clarity: The presentation is clear, and the ideas are easy to follow.
3. Reproducibility: The paper provides the code repo to reproduce the experiments, which is beneficial for future work to build on top of it.

**Weaknesses:**

1. Lack of novelty: the overall design and each of the three components of UniAdapter are not interesting. In particular, using shared weights in the lower layers followed by layers with specialized weights is common in multi-task learning literature. Weight-sharing has also been employed by previous parameter-efficient fine-tuning work like Compacter (Karimi Mahabadi et al., 2021). Using residual connections is again a commonly seen trick. More importantly, the performance improvement that resulted from combining these three techniques is not impressive at all compared with vanilla adapters, as shown in the middle rows of Table 2.

2. Related work: The absence of related work published in 2023 from the first three sections of the paper is surprising. Only a few recent methods are used as baselines in the last experiment section.

3. Comparison fairness: In Table 2, the highlighted best-performing result is UniAdapter with r=512, which uses 19.0M parameters, significantly more than the middle rows. The comparison is kind of unfair, and it would be better to include "Adapter with r=512" for a fair evaluation.

4: Scaling: The paper mentions that UniAdapter is currently only integrated with BLIP. It raises the question of how the method scales to larger models, such as BLIP2, SimVLP, BEIT 3. Further investigation into the scalability and applicability of UniAdapter is needed.

**Questions:**

See Weaknesses 2-4.

---

> ### Author Response · Authors · 2023-11-18
> **Response to Reviewer 1KDb (Part 1/3)**
>
> We would like to begin by expressing our sincere gratitude for your thorough review of our paper. We greatly appreciate your suggestions, which are crucial in improving the quality of our paper. The questions you raised are insightful, which we believe have been carefully clarified and addressed as follows.
>
> **Q1: The overall design and each of the three components of UniAdapter are not interesting.**
>
> > **(1) The overall design and each of the three components of UniAdapter are not interesting. In particular, using shared weights in the lower layers followed by layers with specialized weights is common in multi-task learning literature. Weight-sharing has also been employed by previous parameter-efficient fine-tuning work like Compacter (Karimi Mahabadi et al., 2021).**
>
> **A:** Thank you for your comments regarding the design and components of UniAdapter, particularly about the use of parameter-sharing, a common strategy in multi-task learning and parameter-efficient fine-tuning as seen in works like Compacter.
>
> We acknowledge that parameter-sharing is not a novel concept per se, but its effective application in different contexts is key. In contrast to previous works where weight-sharing mainly supports multi-task learning, our approach utilizes it to enhance multimodal transfer learning, specifically aiming at the fusion of different modalities within the adapter.
>
> * Although previous works (e.g., Compacter) utilize the concept of weight-sharing, the motivations and goals of the two models are different. Most multi-task work aimed to make the model perform well on multi-tasks, thus opting for weight sharing at low layers, and they have achieved impressive performances on a wide range of benchmarks.  In UniAdapter, we leverage parameter-sharing in the down-projection layer of the UniAdapter the nuanced similarities and differences between modalities, which bring better performance with fewer data. Meanwhile, the design of weight-sharing in UniAdapter is at adapter level, will previous multi-task methods directly shared weights in the lower layers.
> * To provide deeper insight, we have conducted additional visualizations (**see Figure4 and Figure5 in the Appendix part**). We select input representations from our UniAdapter at the 12th layer originating from different modality modules. After that, we conduct joint t-SNE and PCA visualizations on these representations. Our experimental data is derived from 128 randomly selected images from Flickr30K. The findings are as follows:
>     - Inputs representations to the UniAdapter from various modalities exhibit similar spatial distributions, suggesting that they reside within a shared space (or overlapped spaces across distributions). This supports the effectiveness of our down-projection sharing strategies.
>     - Further, we compared the input representations of the down-projection layer with those of the up-projection layer. Here, we noted that different modalities' data distributions occupy distinct spaces when it comes to the up-projection layer. This finding validates our experiments that sharing the up-projection layer negatively impact the model's performance, as it appears to interfere with the distinct representation spaces required for different modalities.
>
>    - We hope the underlying insight behind this approach will inspire future works in this domain.
> * We also have compared different layers of the modality sharing in **Table 12** in the appendix. The results indicate that **sharing weight for lower layers may lead to worse performance** for multimodal tasks. Our findings are different from multi-task learning methods (e.g., using shared weights in the lower layers).
> * Our parameter-sharing mechanism in UniAdapter is meticulously designed, encompassing not only down-projection but also query residual adapters, query-redisual up-projection layers, and encoder-decoder models for tasks like Visual Question Answering (VQA). This comprehensive approach is specifically engineered for cross-modal parameter-efficient transfer learning, addressing unique challenges in this domain.
>
> > **(2) Using residual connections is again a commonly seen trick.**
>
> **A:** Thank you for noting the widespread use of residual connections in the field. We agree that employing residual connections is a known technique. However, we would like to emphasize that the contribution of our work is not rooted in the mere utilization of residual connections but in the strategic insights we have developed regarding their implementation: **Language Query information progressively diminishes in the cross-attention arthicture**. In response to this finding, we have strategically incorporated residual connections to mitigate this loss of information. This methodological choice was a targeted response to the specific challenge we identified through our study. This approach has notably improved our model's performance, illustrating the effectiveness.

---

> ### Author Response · Authors · 2023-11-18
> **Response to Reviewer 1KDb (Part 2/3)**
>
> > **(3) More importantly, the performance improvement that resulted from combining these three techniques is not impressive at all compared with vanilla adapters, as shown in the middle rows of Table 2.**
>
> **A:** Thank you for your critique regarding the performance improvements of our UniAdapter, as compared to vanilla adapters in Table 2. We appreciate your perspective and would like to address a potential misunderstanding about the expectations in the realm of parameter-efficient transfer learning, the category under which our UniAdapter method falls.
>
> * In parameter-efficient transfer learning, it's generally recognized that the upper limit of performance is often set by full fine-tuning. As a result, significant performance variances between different adapter models may not be immediately evident.
> * Meanwhile, it's important to note that our UniAdapter achieves significant improvements in specific tasks. For instance, in video retrieval tasks, UniAdapter shows an improvement of 2.5 in Recall@1 **(52.1 vs. 49.6)** compared to the standard Adapter, as shown in Table2. Similarly, in Video VQA tasks, it outperforms by 1.7 **(44.5 vs. 42.8)**. These improvements, while appearing modest, are significant in the context of parameter-efficient transfer learning.
> * More importantly, the UniAdapter not only competes closely with full fine-tuning but, in certain instances, even surpasses it. This is a significant feat, as it challenges the traditional performance benchmarks in this field. Although the numerical increases in performance might seem modest, they represent a substantial leap forward, highlighting the effectiveness and efficiency of our UniAdapter.
>
>
>
> **Q2: Related work: The absence of related work published in 2023 from the first three sections of the paper is surprising. Only a few recent methods are used as baselines in the last experiment section.**
>
>
> **A:** Thank you for pointing out the need to incorporate more recent related work from 2023 in the initial sections of our paper. We value your constructive feedback and recognize the importance of reflecting the latest advancements and trends in our field.
>
> In response to your feedback, we have carefully reviewed and included additional related work published in 2023 in the first three sections of our manuscript.
> Furthermore, we have expanded the set of baselines in the last experiment section to encompass more recent methods.
>
>
> Thank you again for your constructive suggestion regarding the inclusion of more recent related work in the initial sections of our paper. Should there be any specific recent publications that we may have overlooked or that you believe should be included in our manuscript, please feel free to let us know.
>
> **Q3: Comparison fairness: In Table 2, the highlighted best-performing result is UniAdapter with r=512, which uses 19.0M parameters, significantly more than the middle rows. The comparison is kind of unfair, and it would be better to include "Adapter with r=512" for a fair evaluation.**
>
> **A:** We apologize for any confusion caused by the presentation of our results in Table 2. Thank you for bringing this to our attention.
>
> * We would like to clarify that the inclusion of UniAdapter with r=512 in Table 2 was primarily for **the purpose of comparing different configurations of UniAdapter itself** (specifically, r=128 vs. r=512) as well as **full finetuning baseline**. This was not intended as a direct comparison with the standard Adapter model. We realize now that this might not have been clearly conveyed in our original presentation.
>
> * Acknowledging your valid point about ensuring a fair comparison, we have revised Table 2 to include results for the Adapter model with r=512. This adjustment ensures a more balanced and comprehensive evaluation, allowing for a direct comparison of UniAdapter's performance with that of an equivalently parameterized Adapter model.

---

> ### Author Response · Authors · 2023-11-18
> **Response to Reviewer 1KDb (Part 3/3)**
>
> **Q4: Scaling: The paper mentions that UniAdapter is currently only integrated with BLIP. It raises the question of how the method scales to larger models, such as BLIP2, SimVLP, BEIT 3. Further investigation into the scalability and applicability of UniAdapter is needed.**
>
>
> **A:** Thank you for your valuable suggestion regarding the scalability and applicability of UniAdapter to larger models. We acknowledge the importance of demonstrating our method's effectiveness beyond its integration with BLIP.
>
> To address this, we have conducted additional experiments on ALBEF and BLIP-Large in the appendix of our paper (Image-text Retrieval Task on the Flickr30K dataset).  Following your valuable suggestion, we further conduct experments on larger backbone (BEIT 3-Large), the results of these experiments are as follows:
>
>
> | Method                        | I2T  | I2T   | I2T   | T2I  | T2I  | T2I  |
> |-------------------------------|------|-------|-------|------|------|------|
> |                               | R@1  | R@5   | R@10  | R@1  | R@5  | R@10 |
> | ALBEF (Full finetuning)       | 95.9 | 99.8  | 100.0 | 85.6 | 97.5 | 98.9 |
> | UniAdapter (Frozen backbone)  | 95.6 | 99.9  | 100.0 | 84.6 | 97.3 | 98.6 |
> | BLIP-Large (Full finetuning)  | 97.4 | 99.8  | 99.9  | 87.6 | 97.7 | 99.0 |
> | UniAdapter (Frozen backbone)  | 96.9 | 99.9  | 100.0 | 87.5 | 97.9 | 99.1 |
> | BEIT 3-Large (Full finetuning) | 97.1 | 100.0 | 100.0 | 88.0 | 98.0 | 99.0 |
> | UniAdapter (Frozen backbone)  | 96.3 | 99.9  | 100.0 | 87.6 | 97.9 | 99.2 |
>
> These findings indicate that UniAdapter exhibits performance that is either close to or even surpasses full fine-tuning across different backbone models. This not only reinforces the effectiveness of UniAdapter but also demonstrates its scalability and applicability to a broader range of models.
>
> We wish that our response has addressed your concerns, and turns your assessment to the positive side. If you have any questions, please feel free to let us know during the rebuttal window. We appreciate your suggestions and comments! Thank you!

---

> ### Author Response · Authors · 2023-11-21
> **Looking forward to your post-rebuttal feedback**
>
> Dear Reviewer 1KDb,
>
> Thanks again for your insightful suggestions and comments. As the deadline for discussion is approaching, we are happy to provide any additional clarifications that you may need.
>
> In our previous response, we have carefully studied your comments and made detailed responses summarized below:
>
>
> 1. **Clarification on Weight-Sharing Strategy**: We have elaborated on the differences in motivation, experimental methodologies and findings between the weight-sharing strategy used in multi-task learning and our approach for multimodal learning.
> 2. **Performance Improvement Insights**: We've provided a more detailed explanation of the seemingly modest performance improvements, offering deeper insights into their significance.
> 3. **Updated Literature Review**: The first three sections of our paper now include more related works published in 2023, ensuring our research context is current and comprehensive.
> 4. **Fairness in Comparative Analysis**: We've revised Table 2 for greater clarity, ensuring that the comparisons made are fair and transparent.
> 5. **Scalability of UniAdapter**: We've extended our UniAdapter framework to larger backbones (BEIT 3) to demonstrate its scalability and wider applicability.
>
>
>
> We hope that the provided new experiments and additional explanations have convinced you of the merits of our submission.
>
> Please do not hesitate to contact us if there are other clarifications or experiments we can offer. Thanks!
>
> Thank you for your time!
>
> Best, \
> Authors

---

> ### Author Response · Authors · 2023-11-22
> **Thank you and we are looking forward to your post-rebuttal feedback! - one day left for the author-reviewer discussion**
>
> Dear Reviewer 1KDb,
>
> Thank you once again for your insightful suggestions to enhance our work. In response to your concerns about the contribution and weight-sharing, performance improvement, additional related work, and scalability for larger backbones, we **have conducted further experiments and analyses, and have revised our paper accordingly**.
>
> As the rebuttal period is ending soon, we wonder if our response answers your questions and addresses your concerns. Thanks again for your very constructive and insightful feedback!
>
> Best regards, \
> Authors

---

> > ### Comment · Reviewer_1KDb · 2023-11-22
> >
> > I would like to thank the authors for providing their response. I would maintain my score as they only moderately address some of my concerns regarding novelty and effectiveness.

---

> > > ### Author Response · Authors · 2023-11-23
> > >
> > > Dear Reviewer 1KDb,
> > >
> > > We truly appreciate your feedback and are glad to have addressed some of your concerns regarding our paper. We respect your decision to maintain your score and would like to offer additional clarifications on the remaining points about the novelty and effectiveness of our work.
> > >
> > > Thanks for your time and efforts again!
> > >
> > > Best, \
> > > Authors

---

### Official Review · Reviewer_BCaT · 2023-11-03

**Soundness:** 3 good
**Presentation:** 3 good
**Contribution:** 3 good
**Rating:** 6
**Confidence:** 2

**Summary:**

The paper introduces UniAdapter for efficiently transferring vision-language pre-trained models to various cross-modal downstream tasks like video-text retrieval, image-text retrieval, and video and visual question answering. UniAdapter adopts adapters for different modalities (image and video) and tasks while sharing knowledge through partial weight-sharing strategies. UniAdapter outperforms existing state-of-the-art methods and surpasses several full fine-tuning approaches.

**Strengths:**

1. The paper addresses the challenge of unified parameter-efficient cross-modal transfer learning, enabling the efficient use of a pre-trained vision-language model across a variety of cross-modal downstream tasks.

2. The proposed method UniAdapter, a novel framework designed for efficient adaptation, manages to feature a unified adapter architecture that allows for significant parameter efficiency while maintaining or improving task performance.

3. Extensive testing on various cross-modal benchmarks where UniAdapter demonstrated superior performance with fewer parameters compared to previous models.

4. The authors have made the code and models publicly available, promoting transparency and facilitating replication and further research.

**Weaknesses:**

1. While parameter sharing shows advantages in terms of the number of parameters, in reality, the extra parameter count may not be a significant issue. Although the authors have compared the time and memory usage with full fine-tuning, it is uncertain whether this method would retain its advantages if other comparative methods were scaled up in computational resources without regard for parameter amount.

2. The adapter mainly implements some reuse design for multimodal tasks, with its structure not deviating significantly from classical approaches. It is unclear if this is optimal for cross-modal applications. Has the author explored distinct design strategies for different modalities?

3. The method is based on BLIP-base, suggesting potential limitations in the types of models to which it can be applied. Has the author attempted to validate the approach on alternative backbones?

4. The experimental design appears to be somewhat disorganized; it is challenging to discern controlled variables in the comparative analysis presented in each table. This lack of clarity complicates the evaluation of the actual impact of different components of the method.

**Questions:**

Please refer to the weakness.

---

> ### Author Response · Authors · 2023-11-18
> **Response to Reviewer BCaT (Part 1/2)**
>
> Thank you for the positive comments and insightful suggestions. Your insightful questions and valuable suggestions have been immensely helpful in enhancing the paper's quality.
>
> **Q1: While parameter sharing shows advantages in terms of the number of parameters, in reality, the extra parameter count may not be a significant issue. Although the authors have compared the time and memory usage with full fine-tuning, it is uncertain whether this method would retain its advantages if other comparative methods were scaled up in computational resources without regard for parameter amount.**
>
> **A:** Good question! Recent studies, such as LoRA, emphasize that most downstream tasks tend to update parameters **within a low-rank space**. This is a critical observation because adding too many parameters might lead the model to learn more from the downstream task than is actually beneficial, potentially leading to overfitting or diminishing returns. To substantiate this, we conducted experiments on the hidden size of Adapters:
>
> | R@Mean     | R=128 | R=256 | R=768 | R=1024 | R=2048 |
> |------------|-------|-------|-------|--------|--------|
> | Adapter    |    67.4   |   68.2    |    69.4   |   69.2     |    68.5    |
> | UniAdapter |   69.3    |   70.4    |    71.5   |    71.3    |    70.6    |
>
> The results reveal that the performance of Adapters does not consistently improve with an increase in parameter count. And a larger number of parameters does not necessarily equate to better performance and may even lead to worse outcomes. This observation highlights the importance of achieving optimal performance with a judicious use of parameters, which aligns with the core objectives of our research.
>
>
>
>
>
>
> **Q2: The adapter mainly implements some reuse design for multimodal tasks, with its structure not deviating significantly from classical approaches. It is unclear if this is optimal for cross-modal applications. Has the author explored distinct design strategies for different modalities?**
>
> **A:** Thank you for your thoughtful question about our design strategy for different modalities in multimodal tasks.
>
> * We have indeed explored unique adapter designs for varying modalities. Specifically, for video tasks, we experimented a video-specific adapter (ST-Adapter [1]), for the visual backbone. Simultaneously, we used Parallel Adapters for the textual and multimodal backbones. Despite this modality-specific approach, our results, detailed in Table 4, reveal that the ST-Adapter was marginally less effective than employing Parallel Adapters across all modalities. We observed a slight performance difference (69.7 vs. 69.6 on R@Mean) while achieving a significant cost in parameter count (from 42.5M to 28.4M).
>
> * Meanwhile, our findings in Table 1 highlight that for multimodal tasks, the integration of adapters in the cross-modal modality is more impactful than in the visual or textual modalities. This insight is pivotal to our UniAdapter design, where we focus on optimizing the multimodal aspect. Consequently, we have adopted a strategy of minimizing parameter count in the visual and textual modalities through parameter sharing. This approach avoids over-specialization in either modality, aligning with our goal of a more streamlined and efficient design.
>
> * Our investigations suggest that a more uniform module design across different modalities tends to yield superior results compared to the introduction of modality-specific, specialized designs. This finding supports the notion that for cross-modal applications, a unified approach to adapter design can be more effective, as opposed to distinct, modality-specific implementations.

---

> ### Author Response · Authors · 2023-11-18
> **Response to Reviewer BCaT (Part 2/2)**
>
> **Q3: The method is based on BLIP-base, suggesting potential limitations in the types of models to which it can be applied. Has the author attempted to validate the approach on alternative backbones?**
>
> **A:** Thank you for your valuable suggestion regarding the scalability and applicability of UniAdapter to alternative backbones. We acknowledge the importance of demonstrating our method's effectiveness beyond its integration with BLIP-base.
>
> To address this, we have conducted additional experiments on ALBEF and BLIP-Large in **Table 11 and Table12 in Appendix of our paper** (Image-text Retrieval Task on the Flickr30K dataset, now merged in Table 11).  Following your valuable suggestion, we further conduct experments on larger backbone (BEIT 3-Large), the results of these experiments are as follows:
>
>
> | Method                        | I2T  | I2T   | I2T   | T2I  | T2I  | T2I  |
> |-------------------------------|------|-------|-------|------|------|------|
> |                               | R@1  | R@5   | R@10  | R@1  | R@5  | R@10 |
> | ALBEF (Full finetuning)       | 95.9 | 99.8  | 100.0 | 85.6 | 97.5 | 98.9 |
> | UniAdapter (Frozen backbone)  | 95.6 | 99.9  | 100.0 | 84.6 | 97.3 | 98.6 |
> | BLIP-Large (Full finetuning)  | 97.4 | 99.8  | 99.9  | 87.6 | 97.7 | 99.0 |
> | UniAdapter (Frozen backbone)  | 96.9 | 99.9  | 100.0 | 87.5 | 97.9 | 99.1 |
> | BEIT 3-Large (Full finetuning) | 97.1 | 100.0 | 100.0 | 88.0 | 98.0 | 99.0 |
> | UniAdapter (Frozen backbone)  | 96.3 | 99.9  | 100.0 | 87.6 | 97.9 | 99.2 |
>
> These findings indicate that UniAdapter exhibits performance that is either close to or even surpasses full fine-tuning across different backbone models. This not only reinforces the effectiveness of UniAdapter but also demonstrates its scalability and applicability to a broader range of models.
>
>
>
> **Q4: The experimental design appears to be somewhat disorganized; it is challenging to discern controlled variables in the comparative analysis presented in each table. This lack of clarity complicates the evaluation of the actual impact of different components of the method.**
>
> **A:** Thank you for your feedback on the organization of our experimental design, we acknowledge that the presentation in Table 2 might have led to some confusion.
>
> In response to your valuable input, we have revised Table 2 to include results for r=512. This update ensures that all ablation studies are consistently conducted at r=512. By doing so, we aim to provide a more coherent and straightforward comparison across different experimental setups. We believe that this adjustment will significantly improve the clarity of our experimental design and thank you for your suggestion in making our research more accessible and comprehensible.
>
> [1] Pan, J., Lin, Z., Zhu, X., Shao, J., & Li, H. (2022). St-adapter: Parameter-efficient image-to-video transfer learning. Advances in Neural Information Processing Systems, 35, 26462-26477.
>
> Thanks again for your time and effort! For any other questions, please feel free to let us know during the rebuttal window.

---

### Official Review · Reviewer_euaM · 2023-11-10

**Soundness:** 3 good
**Presentation:** 3 good
**Contribution:** 2 fair
**Rating:** 6
**Confidence:** 4

**Summary:**

This paper introduces a novel and parameter-efficient adapter that facilitates the transfer of pretrained knowledge to various vision-language downstream tasks. The experimental results demonstrate the remarkable effectiveness of this approach.

**Strengths:**

1. The Uni-adapter incorporates partly shared layers, leading to a reduction in trainable parameters while simultaneously improving performance.

2. The experiments conducted in this paper encompass a wide range of common datasets for downstream tasks, illustrating the generalization and effectiveness of the proposed method.

**Weaknesses:**

1. Some of the ideas presented in this paper have been explored in previous works. For example, (1) the sharing of layers across multiple modalities has been addressed in [1]; (2) the aggregation of video (frame) features in a parameter-free manner, such as through attention or averaging, has also been discussed in [2].

2. Regarding feature visualization, I suggest conducting a comparison between the results of the non-shared architecture, the up-shared one, and the all-shared one.

Ref: [1] Image as a Foreign Language: BEIT Pretraining for All Vision and Vision-Language Tasks. https://arxiv.org/pdf/2208.10442.pdf [2] CLIP4Clip: An Empirical Study of CLIP for End to End Video Clip Retrieval. https://arxiv.org/pdf/2104.08860.pdf

**Questions:**

Please refer to weaknesses.

---

> ### Author Response · Authors · 2023-11-18
> **Response to Reviewer euaM**
>
> Thank you for the positive comments and insightful suggestions. Your insightful questions and valuable suggestions have been immensely helpful in enhancing the paper's quality.
>
>
> **Q1: Some of the ideas presented in this paper have been explored in previous works.**
>
> > **(1) the sharing of layers across multiple modalities has been addressed in BEIT-3**
>
> **A:** Parameter-sharing is a commonly used strategy, but how to correctly applied it is much more important. Although full parameter-sharing methods have achieve great success, recent arts (e.g., BLIP-2) show that proper partial parameter-sharing is more important.
> 1. In this paper, we have carefully designed our parameter-sharing approach, considering not only down-projection parameter-sharing but also parameter-sharing for query residual adapters, up-projection adapters, and encoder-decoder models on VQA tasks. Our approach is tailored specifically for cross-modal parameter-efficient transfer learning.
> 2. Regarding the comparison with BEIT-3, it's important to note that BEIT-3 employs direct Multi-Head Self-Attention sharing for each modality, a standard practice. In contrast, our UniAdapter sharing strategy is custom-designed for efficient cross-modal Adapter modules.
> 3. Meanwhile, designing an effective parameter-sharing module for parameter-efficient cross-modal modeling is non-trivial and sometimes even "counter-intuition". Our proposed strategy is carefully designed to meet the demands of cross-modal parameter-efficient transfer learning, and it is not a direct increment of previous methods.
>
> We believe that our work will inspire and assist researchers in designing better models for this task.
>
>
> > **(2) the aggregation of video (frame) features in a parameter-free manner, such as through attention or averaging, has also been discussed in CLIP4Clip.**
>
> **A:** Parameter-free aggregation of video (frame) features has indeed been discussed before. Nevertheless, how to better perform aggregation is even more important. In CLIP4Clip, the authors explored different schemes (including average, seqLSTM, seqTransf, tightTransf). However, they found that a simple average surpassed other careful designs, suggesting that parameter-free aggregation is a challenging task.
>
> Our UniAdapter employs language-guided free aggregation. Without adding extra parameters and with minimal computational overhead, it yields improved results. Additionally, we offer comparison results with various parameter-free aggregation strategies in the table below.
>
> | Didemo       | R@1  | R@5  | R@10 | R@Mean |
> |-|------|-|-|-|
> | Average      | 47.4 | 70.1 | 79.7 | 65.7   |
> | Token-Concat (our Baseline) | 51.0 | 75.8 | 84.2 | 70.3   |
> | PFA (ours)   | **52.1** | **77.3** | **85.2** | **71.5**   |
>
> Meanwhile, it is important to note that parameter-free aggregation (PFA) is **not** the core contribution of this work. Our UniAdapter mainly aims to construct a unified parameter-efficient training framework designed for multimodal tasks.
>
> **Q2: Regarding feature visualization, I suggest conducting a comparison between the results of the non-shared architecture, the up-shared one, and the all-shared one.**
>
> **A:** Thank you for your valuable suggestion to conduct a comparative analysis of non-shared, up-shared, and all-shared architectures within our UniAdapter model. Following your recommendation, we have conducted an in-depth comparison, the results of which are now included in **Figure 4 and 5 in the Appendix**.
>
> 1. We observed that input representations to the UniAdapter from various sharing-strategy exhibit similar spatial distributions. This finding suggests that these representations reside within a shared space or have overlapped distributions across different modalities. Interestingly, the type of adapter employed (e.lg, non-shared vs. shared) does not seem to significantly impact this distribution, which we attribute to the majority of the model's parameters being frozen. This observation supports the effectiveness of our down-projection sharing strategies.
> 2. Further, we compared the input representations of the down-projection layer with those of the up-projection layer. Here, we noted that different modalities' data distributions occupy distinct spaces when it comes to the up-projection layer. This finding validates our experiments that sharing the up-projection layer negatively impact the model's performance, as it appears to interfere with the distinct representation spaces required for different modalities.
>
> These additional visualizations and analyses provide valuable insights into the distribution of multimodal data within our UniAdapter model and underscore the impact of our architectural choices on the model's performance. We believe these findings significantly contribute to understanding the nuances of our UniAdapter's design and functionality.
>
>
> Thanks again for your time and effort! For any other questions, please feel free to let us know during the rebuttal window.

---

> > ### Comment · Reviewer_euaM · 2023-11-22
> >
> > Dear Author,
> >
> > Thank you for your detailed rebuttal and the effort put into addressing the concerns raised during the review. After careful consideration of your responses and a re-evaluation of the manuscript, I have decided to maintain my original score.

---

> ### Author Response · Authors · 2023-11-22
>
> Dear Reviewer euaM,
>
> Thank you once more for dedicating a significant amount of time to review our paper. Your insights have greatly contributed to enhancing the quality and clarity of our work. We are pleased to observe that our response has effectively addressed the majority of your concerns.
>
> We sincerely appreciate your continued time and effort.
>
> Warm regards, \
> Authors

---

### Author Response · Authors · 2023-11-18
**General Response**

We sincerely appreciate all reviewers’ time and efforts in reviewing our paper. We are glad to find that reviewers generally recognized our contributions:

* **Model.** A novel framework designed for efficient adaptation [BCaT], manages to feature a unified adapter architecture that allows for significant parameter efficiency while maintaining or improving task performance [2kKp, BCaT, euaM] . The residual part for both inside adapter and in multi-modal encoder parts are great ideas to apply residual learning for text [2kKp]. The problem of using a unified adapter architecture (and potentially shared weights) for modeling single-modal and multi-modal interactions is interesting [1KDb].
* **Experiment.** Extensive testing on various cross-modal benchmarks where UniAdapter demonstrated superior performance with fewer parameters compared to previous models [BCaT, 1KDb, 2kKp, euaM].
* **Writing.** Clear and nice writing. It is easy to understand its concept and contributions [2kKp, 1KDb], the ideas are easy to follow [1KDb].
* **Code.** The authors have made the code and models publicly available, promoting transparency and facilitating replication and further research [BCaT, 1KDb, 2kKp].


And we also thank all reviewers for their insightful and constructive suggestions, which help a lot in further improving our paper. In addition to the pointwise responses below, we summarize supporting experiments added in the rebuttal according to reviewers’ suggestions.

**New Experiments.**
* Expanding the hidden size of Adapters [BCaT].
* Utilizing our UniAdapter with larger backbones [BCaT, 1KDb].
* Presenting visualization results for the PFA mechanism [2kKp].
* Extending t-SNE visualizations of UniAdapter representations [euaM].
* Comparing results for video frame aggregation [euaM].
* Comparing results under different text conditions for the PFA mechanism [2kKp].

We thank all reviewers for their insightful and constructive suggestions, which help a lot in further improving our paper. We hope that our pointwise responses below could clarify all reviewers’ confusion and alleviate all of their concerns. We thank all reviewers’ time again and we are always ready to solve your concerns.

Best, \
Authors

---

> ### Author Response · Authors · 2023-11-23
> **Response to Remaining Concerns in Rebuttal**
>
> Firstly, we would like to express our gratitude for all reviewer feedback and for recognizing the contributions of our paper. We appreciate the time and effort.
>
> In the rebuttal, we addressed most of the reviewers' concerns. However, we noticed that several concerns remain, which we will elaborate on here.
>
>
> ## Novelty && Contribution:
>
> * We understand that some designs in our UniAdapter may not be considered groundbreaking. However, our primary objective is to explore a new paradigm for simple yet effective parameter-efficient training in the multimodal field and establish a unified framework. These design explorations include the appropriate location, parameters-sharing, residual-query adaptation, and more. We have provided rich ablation studies and made our code open-sourced.
>
> * Moreover, our design is straightforward and effective without intricate modules, we believe it's unnecessary to inject excessively complex designs to artificially make it "looks novel". In contrast, it is always desirable that a simple method can finish the task well, especially given that our main novelty lies in the paradigm and subtle designs.
>
> * We sincerely hope that our work can serve as an inspiration and offer a valuable starting point for future research in this field.
>
>
> Sincerely, \
> Authors

---

### Meta-Review · Area_Chair_yfoe · 2023-12-04

**Metareview:**

This work develops a new parameter-efficient tuning method for VLM models. Although the parameter efficient tuning methods have been widely explored for language models and vision models, similar approaches for large-scale VLM models are rare. This work proposes a timely solution. Specifically, the authors introduce the weight-sharing adapter modules across different modalities. Comprehensive experiments demonstrate effectiveness of the proposed method.

**Justification For Why Not Higher Score:**

- The idea is not new. The adapter methods have been widely explored for language and vision models. Weight sharing has been explored in multi-task learning.

**Justification For Why Not Lower Score:**

- The proposed solution is timely. Efficiently fine-tuning a large VLM model is necessary and the demand is increasing.
- The method is simple and effective.

---

### Decision · Program_Chairs · 2024-01-16

Accept (poster)